# OpenGSL: A Comprehensive Benchmark for Graph Structure Learning

**Zhiyao Zhou**[1], **Sheng Zhou**[2],[*] **Bochao Mao**[2], **Xuanyi Zhou**[1], **Jiawei Chen**[1]
**Qiaoyu Tan**[3], **Daochen Zha**[4], **Yan Feng**[1], **Chun Chen**[1], **Can Wang**[1]
[1]College of Computer Science, Zhejiang University, Hangzhou, China
[2]School of Software Technology, Zhejiang University, Ningbo, China
[3]New York University Shanghai [4]Rice University
{zjucszzy, zhousheng_zju, bcmao, zxy2004
sleepyhunt, fengyan, chenc, wcan}@zju.edu.cn
qiaoyu.tan@nyu.edu  daochen.zha@rice.edu

## Abstract

Graph Neural Networks (GNNs) have emerged as the *de facto* standard for representation learning on graphs, owing to their ability to effectively integrate graph topology and node attributes. However, the inherent suboptimal nature of node connections, resulting from the complex and contingent formation process of graphs, presents significant challenges in modeling them effectively. To tackle this issue, Graph Structure Learning (GSL), a family of data-centric learning approaches, has garnered substantial attention in recent years. The core concept behind GSL is to jointly optimize the graph structure and the corresponding GNN models. Despite the proposal of numerous GSL methods, the progress in this field remains unclear due to inconsistent experimental protocols, including variations in datasets, data processing techniques, and splitting strategies. In this paper, we introduce OpenGSL, the first comprehensive benchmark for GSL, aimed at addressing this gap. OpenGSL enables a fair comparison among state-of-the-art GSL methods by evaluating them across various popular datasets using uniform data processing and splitting strategies. Through extensive experiments, we observe that existing GSL methods do not consistently outperform vanilla GNN counterparts. We also find that there is no significant correlation between the homophily of the learned structure and task performance, challenging the common belief. Moreover, we observe that the learned graph structure demonstrates a strong generalization ability across different GNN models, despite the high computational and space consumption. We hope that our open-sourced library will facilitate rapid and equitable evaluation and inspire further innovative research in this field. The code of the benchmark can be found in `https://github.com/OpenGSL/OpenGSL`.

## 1 Introduction

Graph Neural Networks (GNNs) [6, 18, 13, 41] have emerged as the dominant approach for learning on graph-structured data, thanks to their exceptional ability to leverage both the graph topology structure and node attributes [44]. Considerable endeavors have been recently made to enhance the performance of GNNs by refining the architectures of GNN models, such as neural message passing [41, 46, 45, 11, 3] and transformer based methods [19, 48, 35]. However, these *model-centric* methods overlook the potential flaws of the underlying graph structure, which can result in suboptimal performance. Notably, extensive evidence from previous studies [5] confirms that real-world

---

[*]Corresponding Author

37th Conference on Neural Information Processing Systems (NeurIPS 2023) Track on Datasets and Benchmarks.

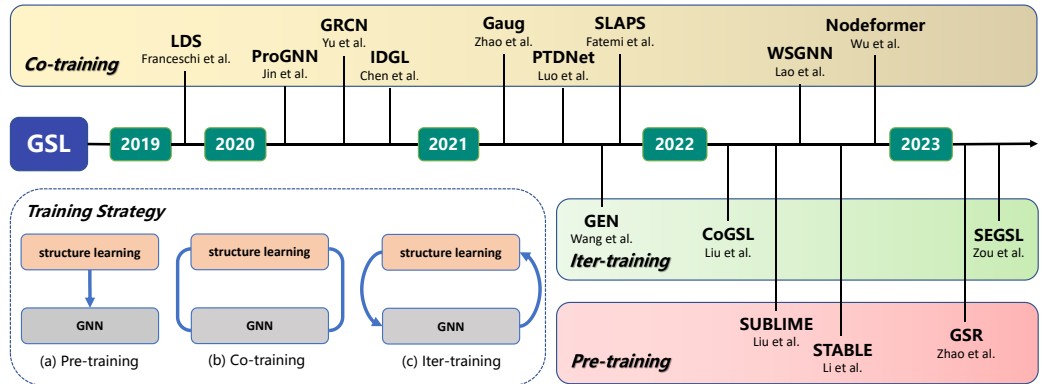

Figure 1: Timeline of GSL research. Existing GSL methods are categorized into three groups based on the training procedure. Bottom left corner illustrates the key difference on the interaction between two components.

graphs often exhibit suboptimal characteristics, such as the absence of valuable links and the presence of spurious connections among nodes.

To improve the graph quality, Graph Structure Learning (GSL) [58], a family of *data-centric* graph learning methods [51, 50, 30], has recently attracted considerable research interest. These methods target optimizing the graph structure and the corresponding GNN representations jointly [26, 22, 16, 9, 42]. By refining the graph structure, GSL methods can potentially empower GNNs to learn better representations with improved performance. GSL has been successfully applied to disease analysis [4] and protein structure prediction [17].

Despite the plethora of GSL methods proposed in recent years, as illustrated in Figure 1, there is no comprehensive benchmark for GSL, which significantly impedes the understanding and progress of GSL in several aspects. *i)* The use of different datasets, data processing approaches and data splitting strategies in previous works makes many of the results incomparable. *ii)* There is a lack of understanding of the learned structure itself, particularly regarding its homophily and generalizability to GNN models other than GCN. *iii)* Apart from accuracy, understanding each method's computation and memory costs is imperative, yet often overlooked in the literature.

To bridge this gap, we introduce OpenGSL, the first comprehensive benchmark for GSL. OpenGSL implements a wide range of GSL algorithms through unified APIs, while also adopting consistent data processing and data splitting approaches for fair comparisons. Through benchmarking the existing GSL methods on various datasets, we make the following contributions:

- **Comprehensive benchmark.** OpenGSL enables a fair comparison among thirteen state-of-the-art GSL methods by unifying the experimental settings across ten popular datasets of diverse characteristics. Surprisingly, the empirical results reveal that GSL methods do not consistently outperform the vanilla GNNs on all datasets.
- **Multi-dimensional analysis.** We conduct a systematic analysis of GSL methods from various dimensions, encompassing the homophily of the learned structure, the generalizability of the learned structure across GNN models, and the time and memory efficiency of the existing methods. **Our key findings:** *i)* Contrary to the common belief in the homophily assumption, increasing the homophily of the structure does not necessarily translate into improved performance. *ii)* The learned structures by GSL methods exhibit strong generalizability. *iii)* Most GSL methods are time- and memory-inefficient, some of which require orders of magnitudes more resources than vanilla GNNs, highlighting the pressing need for more efficient GSL approaches.
- **Open-sourced benchmark library and future directions**: We have made our benchmark library publicly available on GitHub, aiming to facilitate future research endeavors. We have also outlined potential future directions based on our benchmark findings to inspire further investigations.

To summarize, in this paper, we aim to create a comprehensive benchmark that facilitates the fair evaluation of graph structure learning algorithms, encourages new research, and ultimately advances the progress of the field as a whole.

## 2 Formulations and Background

**Notations.** Let $\mathcal{G} = (\mathcal{V}, \mathcal{E}, \mathbf{A}, \mathbf{X})$ be a graph, where $\mathcal{V}$ is the set of $N$ nodes and $\mathbf{A} \in \mathbb{R}^{N \times N}$ is the adjacency matrix. $\mathcal{E}$ denotes the edge set and $\mathbf{X} \in \mathbb{R}^{N \times d}$ represents the corresponding feature matrix with dimension $d$. Typically, a GNN model is often parameterized by a mapping function $f : (\mathbf{A}, \mathbf{X}) \to \mathbf{H} \in \mathbb{R}^{N \times l}$, which maps each node $v \in \mathcal{V}$ into a $l$-dimensional embedding vector $\mathbf{h}_v$. In the semi-supervised setting, some nodes $v_i \in \mathcal{V}_{train}$ are often associated with labels $y_i$ to guide the training. Please note that in the traditional GNNs, the adjacent matrix $\mathbf{A}$ only serves as input and is not updated along with the training of GNNs. In the GSL methods, a gradually updated structure $\mathbf{S} \in \mathbb{R}^{N \times N}$ is learned to replace the original structure $\mathbf{A}$ during training process. This is the major difference between traditional GNNs and GSL methods.

**Timeline of GSL methods through the lens of training procedures.** To provide a global understanding of the literature, we provide a high-level overview of the existing GSL methods based on their *training procedure*. Specifically, we identify two key components in GSL methods: structure learning component and GNN component. Based on the interaction between these two components, we partition existing GSL methods into three categories, depicted in bottom left corner of Figure 1: (a) pre-training, (b) co-training, and (c) iter-training. Pre-training involves a two-stage learning process, where the structure is learned through pre-training and then used to train GNNs in downstream tasks [26, 22, 52]. In co-training methods [10, 2, 28], neural networks that generate the graph structure are optimized together with GNNs. The iterative methods [42, 59, 38] involve training the two components iteratively; they learn the structure from predictions or representations generated by an optimized GNN and use it to train a new GNN model for the subsequent iteration. A similar taxonomy can be found in [1]. Our taxonomy differs by introducing the pre-training category and combining two categories (named joint learning and adaptive learning in [1]) into one named co-training.

**Homophily and Heterophily.** Homophily [31] and heterophily are two mutually exclusive measurements based on similarity between connected node pairs, where two nodes are considered similar if they share the same node label. The homophily of graph $\mathcal{G}$ can be formulated as:

$$\text{homo}(\mathcal{G}) = \frac{1}{|\mathcal{E}|} |\{(v_i, v_j) | (v_i, v_j) \in \mathcal{E}, y_i = y_j\}| \tag{1}$$

where $|\mathcal{E}|$ is the number of observed edges. Correspondingly, the heterophily of graph $\mathcal{G}$ is defined as $1 - \text{homo}(\mathcal{G})$. The homophily of graphs has been widely assumed to be the key motivation for designing GNNs, while a few recent works [29, 27] have argued on it. Although some GSL methods [53, 42] also claim to learn a graph structure with high homophily, the homophily of the graph structure learned by GSL methods has not been studied.

## 3 Benchmark Design

We begin by introducing the datasets utilized in our benchmarking process, along with the algorithm implementations. Then, we outline the research questions that guide our benchmarking study.

### 3.1 Datasets and Implementations

**Datasets.** In order to provide a comprehensive evaluation of existing GSL methods, we collect 10 graph node classification datasets that have been widely used in the GSL literature. These selected datasets come from different domains and exhibit different characteristics, enabling us to evaluate the generalizability of existing methods across a range of scenarios. Specifically, we use three classic citation datasets [36], namely Cora, Citeseer, Pubmed, as well as two representative social network datasets BlogCatalog and Flickr [15]. Additionally, we include five datasets that have been proposed recently in [34], due to their ability to overcome the drawbacks of the commonly used heterophilous datasets [32]. Table 1 shows the statistics of these datasets, which are divided into two groups according to whether the edge homophily is higher than 0.5. Note that in the five homophilous graph

Table 1: Overview of the datasets used in this study.

| group | Dataset | # Nodes | # Edges | # Feat. | Avg. # degree | # Classes | # Homophily |
|---|---|---|---|---|---|---|---|
| homophilous | Cora | 2,708 | 5,278 | 1,433 | 3.9 | 7 | 0.81 |
| | Citeseer | 3,327 | 4,552 | 3,703 | 2.7 | 6 | 0.74 |
| | Pubmed | 19,717 | 44,324 | 500 | 4.5 | 3 | 0.80 |
| | Questions | 48,921 | 153,540 | 301 | 6.3 | 2 | 0.84 |
| | Minesweeper | 10,000 | 39,402 | 7 | 7.9 | 2 | 0.68 |
| heterophilous | BlogCatalog | 5,196 | 171,743 | 8,189 | 66.1 | 6 | 0.40 |
| | Flickr | 7,575 | 239,738 | 12,047 | 63.3 | 9 | 0.24 |
| | Amazon-ratings | 24,492 | 93,050 | 300 | 7.6 | 5 | 0.38 |
| | Roman-empire | 22,662 | 32,927 | 300 | 2.9 | 18 | 0.05 |
| | Wiki-cooc | 10,000 | 2,243,042 | 100 | 448.6 | 5 | 0.34 |

datasets, Questions and Minesweeper are binary classification datasets with *highly imbalanced class distributions*. We present more detailed introductions in Appendix A.1, along with other common graph statistics and newly proposed metrics [33] that can better capture the characteristics of graph datasets.

The data splitting methods in different GSL works are not consistent, bringing difficulties in conducting fair comparisons. We investigate various GSL works and choose the data splits that are most commonly used. For three citation datasets, we use the classic split from [47, 18]. For BlogCatalog and Flickr, we follow the split in [15, 53]. For five datasets from [34], we follow the original split in [34].

**Implementations.** We consider a collection of state-of-the-art algorithms, including LDS [10], ProGNN [16], IDGL [2], GRCN [49], GAug [53], SLAPS[2] [9], GEN [42], WSGNN [20], Nodeformer [43], CoGSL [24], SUBLIME [26], STABLE [22], and SEGSL [59]. We rigorously reproduced all methods according to their papers and source codes. To ensure a fair evaluation, we perform hyperparameter tuning with the same search budget on the same dataset for all methods. More details on these algorithms and implementations can be found in Appendix A.2.

## 3.2 Research Questions

We carefully design the OpenGSL to systematically evaluate existing methods and inspire future research. Specifically, we aim to answer the following research questions.

**RQ1: How much progress has been made by existing GSL methods?**

**Motivation.** Previous research in GSL has been hindered by the use of different data preprocessing, and splits, which make it difficult to fairly evaluate and compare the performance of different methods. Given the fair comparison environment provided by OpenGSL, the first research question is to revisit how much progress has been made by existing GSL methods. By answering this question, we aim to gain a deeper understanding of the strengths and weaknesses of existing methods, and identify areas that offer potential for further enhancements.

**Experiment Design.** Following the experimental setting of most existing GSL methods, we conduct the node classification experiments on all the datasets. For each method and dataset, we report the mean performance and standard deviation of 10 runs. For binary classification datasets, we report ROC AUC metric, while for other datasets, we report accuracy metric. Since most of the implemented GSL methods have used GCN as backbone, we compare the performances of GSL methods with vanilla GCN to verify the enhancement of learning structures. Additional results using other backbones can be found in Appendix D.5.

**RQ2: Does GSL benefit from learning graph structures with higher homophily?**

**Motivation.** The homophily assumption has been a fundamental motivation of modern GNNs' designs, which has also been brought to the GSL scenarios. More specifically, some existing GSL methods have attempted to learn the structure with higher homophily by introducing explicit

---

[2]SLAPS does not use the original graph structure, thus being worse in most cases. Results of all methods without accessing the graph structure are provided in Appendix D.7.

homophily-oriented objectives [53, 42]. However, these claims are questionable since they have been evaluated only on a limited set of toy datasets [42], with little examination of their validity on real datasets. As researchers have started to question the homophily assumption on GNNs [29], it becomes imperative to re-evaluate the significance of GSL methods in learning more homophilous graph structures.

**Experiment Design.** To answer this question, we first compare the homophily of the structure learned[3] by GSL methods with the original one. Then we determine whether the reported performance improvements stem from a more homophilous graph structure by examining the correlation between the homophily and node classification performance.

**RQ3: Can the learned structures generalize to other GNN models?**

**Motivation.** Node classification tasks have been a popular means of evaluating GNNs jointly optimized with GSL methods, the quality of the learned graph structure, however, has not been thoroughly evaluated. While GSL has demonstrated improvements in certain cases, it remains uncertain whether these improvements can be attributed to learning superior structures. Furthermore, it is unclear whether the learned structures have the capability to generalize effectively to other GNN models. Therefore, it is crucial to conduct experiments to evaluate the generalizability of the learned graph structures.

**Experiment Design.** To answer this research question, we use the learned structure and original features to create a new graph data $\mathcal{G}' = (\mathbf{S}, \mathbf{X})$, and train a new GNN method on the new graph. This differs from the setting in RQ1, as we treat each GSL method as a pre-processing step and train a GNN model from scratch using the learned structure, rather than using a GNN jointly optimized with the structure. By comparing the performance on the original graph and the new graph, we can evaluate the generalizability of the learned structure. To further verify the effectiveness of learned structure, we also include two non-GNN methods, Label Propagation [57, 55] and LINK [54], that only take graph structure as input without using node features for node classification.

**RQ4: Are existing GSL methods efficient in terms of time and space?**

**Motivation.** As the GSL methods simultaneously optimize the GNNs and graph structure, they naturally consume more computational complexity and space than the GNNs. However, the efficiency of GSL methods have been largely overlooked by existing methods. Although introducing the structure learning may benefit the GNNs, the extra computational consumption has posed significant requirements of trade-off between performance and efficiency. It is critical to understand the trade-off for deploying the GSL in the practical applications.

**Experiment Design.** To answer this research question, we evaluate the efficiency of each GSL method in terms of time and space. Specifically, we record the wall clock running time and peak GPU memory consumption during each method's training process. For a fair comparison, all experiments in this part were conducted on a single NVIDIA A800 GPU.

## 4 Experiment Results and Analyses

### 4.1 Performance Comparison (RQ1)

We present the performance of all methods on 10 datasets in Table 2 and Table 3. Below are the key findings from these tables.

① **For homophilous graphs, many GSL methods work well in datasets with balanced classes, while they cannot handle highly imbalanced situations.** Table 2 demonstrates that most GSL methods outperform vanilla GCN on balanced homophilous graphs such as Cora, Citeseer, and Pubmed. Out of the 12 methods tested, 7 methods exceeded GCN on at least two datasets, and 9 methods outperformed GCN on at least one dataset. However, some methods were unable to surpass vanilla GCN, suggesting that structure learning might even degrade GNN performance. This result highlights the need to investigate the general effectiveness of GSL methods further. In contrast, the results were vastly different on datasets such as Questions and Minesweeper, where only a few methods show an advantage over GCN. The imbalanced nature of these datasets limits the power of GSL, indicating that their effectiveness is constrained on such types of data. This fact suggests

---

[3]In our experiments we focus on methods that learn a unique structure for simplicity.

Table 2: Node classification results on Cora, Citeseer, Pubmed, Questions and Minesweeper. Shown is the mean ± s.d. of 10 runs with different random seeds. Highlighted are the top **first**, **second**, and **third** results. "−" denotes out of memory or time limit of 24 hours exceeded.

| Model | Cora | Citeseer | Pubmed | Questions | Minesweeper |
|---|---|---|---|---|---|
| GCN | 81.95 ± 0.62 | 71.34 ± 0.48 | 78.98 ± 0.35 | **75.80 ± 0.51** | **78.28 ± 0.44** |
| LDS | **84.13 ± 0.52** | **75.16 ± 0.43** | – | – | – |
| ProGNN | 80.27 ± 0.48 | 71.35 ± 0.42 | 79.39 ± 0.29 | – | 51.43 ± 2.22 |
| IDGL | **84.19 ± 0.61** | **73.26 ± 0.53** | **82.78 ± 0.44** | 50.00 ± 0.00 | 50.00 ± 0.00 |
| GRCN | **84.61 ± 0.34** | 72.34 ± 0.73 | 79.30 ± 0.34 | **74.50 ± 0.84** | 72.57 ± 0.49 |
| GAug | 83.43 ± 0.53 | 72.79 ± 0.86 | 78.73 ± 0.77 | – | **77.93 ± 0.64** |
| SLAPS | 72.29 ± 1.01 | 70.00 ± 1.29 | 70.96 ± 0.99 | – | 50.89 ± 1.72 |
| WSGNN | 83.66 ± 0.30 | 71.15 ± 1.01 | 79.78 ± 0.35 | – | 67.91 ± 3.11 |
| Nodeformer | 78.81 ± 1.21 | 70.39 ± 2.04 | 78.38 ± 1.94 | **72.61 ± 2.29** | 77.29 ± 1.71 |
| GEN | 81.66 ± 0.91 | **73.21 ± 0.62** | 78.49 ± 3.98 | – | **79.56 ± 1.09** |
| CoGSL | 81.46 ± 0.88 | 72.94 ± 0.71 | 78.38 ± 0.41 | – | – |
| SEGSL | 81.04 ± 1.07 | 71.57 ± 0.40 | 79.26 ± 0.67 | – | – |
| SUBLIME | 83.33 ± 0.73 | 72.44 ± 0.89 | **80.56 ± 1.32** | 67.21 ± 0.99 | 49.93 ± 1.36 |
| STABLE | 83.25 ± 0.86 | 70.99 ± 1.19 | **81.46 ± 0.78** | – | 70.78 ± 0.27 |

Table 3: Node classification results on BlogCatalog, Flickr, Amazon-ratings, Roman-empire and Wiki-cooc. Shown is the mean ± s.d. of 10 runs with different random seeds. Highlighted are the top **first**, **second**, and **third** results. "−" denotes out of memory or time limit of 24 hours exceeded.

| Model | BlogCatalog | Flickr | Amazon-ratings | Roman-empire | Wiki-cooc |
|---|---|---|---|---|---|
| GCN | 76.12 ± 0.42 | 61.60 ± 0.49 | 45.24 ± 0.29 | **70.41 ± 0.47** | **92.03 ± 0.19** |
| LDS | 77.10 ± 0.27 | – | – | – | – |
| ProGNN | 73.38 ± 0.30 | 52.88 ± 0.76 | – | 56.21 ± 0.58 | 89.07 ± 5.59 |
| IDGL | 89.68 ± 0.24 | **86.03 ± 0.25** | 45.87 ± 0.58 | 47.10 ± 0.65 | 90.18 ± 0.27 |
| GRCN | 76.08 ± 0.27 | 59.31 ± 0.46 | **50.06 ± 0.38** | 44.41 ± 0.41 | 90.59 ± 0.37 |
| GAug | 76.92 ± 0.34 | 61.98 ± 0.67 | **48.42 ± 0.39** | 52.74 ± 0.48 | **91.30 ± 0.23** |
| SLAPS | **91.73 ± 0.40** | 83.92 ± 0.63 | 40.97 ± 0.45 | **65.35 ± 0.45** | 89.09 ± 0.54 |
| WSGNN | **92.30 ± 0.32** | **89.90 ± 0.19** | 42.36 ± 1.03 | 57.33 ± 0.69 | 90.10 ± 0.28 |
| Nodeformer | 44.53 ± 22.62 | 67.14 ± 6.77 | 41.33 ± 1.25 | 56.54 ± 3.73 | 54.83 ± 4.43 |
| GEN | 90.48 ± 0.99 | 84.84 ± 0.81 | **49.17 ± 0.68** | – | **91.15 ± 0.49** |
| CoGSL | 83.96 ± 0.54 | 75.10 ± 0.47 | 40.82 ± 0.13 | 46.52 ± 0.48 | – |
| SeGSL | 75.03 ± 0.28 | 60.59 ± 0.54 | – | – | – |
| SUBLIME | **95.29 ± 0.26** | **88.74 ± 0.29** | 44.49 ± 0.30 | **63.93 ± 0.27** | 76.10 ± 1.12 |
| STABLE | 71.84 ± 0.56 | 51.36 ± 1.24 | 48.36 ± 0.21 | 41.00 ± 1.18 | 80.46 ± 2.44 |

that as many real-world graphs are imbalanced, evaluating the effectiveness of GSLs on imbalanced datasets should receive more attention in the future research.

② **GSL methods can be effective on specific heterophilous graphs.** Table 3 reveals that some GSL methods, including IDGL, GAug, GEN, and SUBLIME, have the potential to exceed vanilla GCN on heterophilous graphs such as BlogCatalog, Flickr, and Amazon-ratings. This finding is intriguing because homophily is not well-preserved on these graphs, inspiring us to investigate the correlation between homophily and structure learning. However, the results are different on Roman-empire and Wiki-cooc datasets, where none of the methods demonstrate an improvement over GCN. This observation suggests that some heterophilous datasets such as Roman-empire and Wiki-cooc may have informative structural patterns that current GSL methods targeting homophily undermine. For more in-depth analysis, please refer to Section 4.2.

## 4.2 Homophily in Structure Learning (RQ2)

To analyze how different methods behave in terms of homophily and whether the performance is correlated with homophily of the learned structure, we plot the homophily of the learned structure and the node classification performance in the same figure. Figure 2 and Figure 3 show the results, from which we have the following observations:

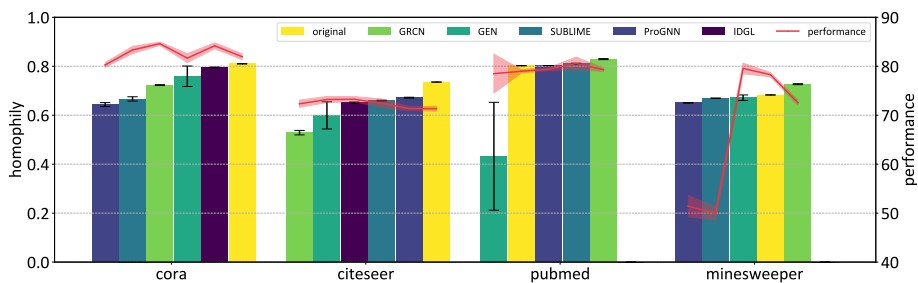

Figure 2: Homophily of learned structures and performances on homophilous datasets. The methods are ordered by homophily.

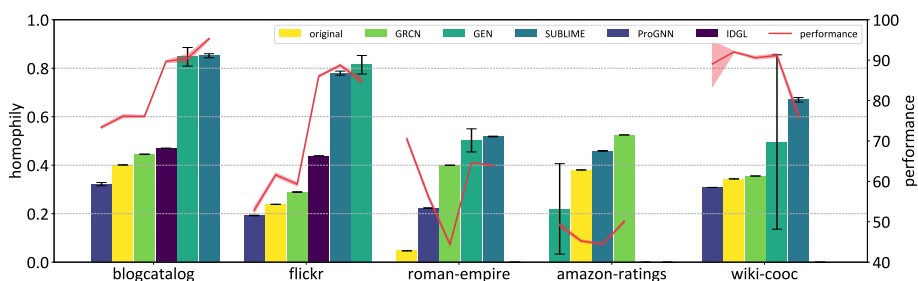

Figure 3: Homophily of learned structures and performances on heterophilous datasets. The methods are ordered by homophily.

Table 4: Pearson's correlation coefficient between homophily of learned structures and accuracy for different methods on various datasets.

| Datasets | Cora | Citeseer | Pubmed | Minesweeper | BlogCatalog | Flickr | Roman-empire | Amazon-ratings | Wiki-cooc |
|---|---|---|---|---|---|---|---|---|---|
| Pearson | 0.29 | -0.50 | 0.62 | 0.50 | 0.86 | 0.87 | -0.25 | -0.11 | -0.84 |

③ **The homophily of the learned structures varies on homophilous and heterophilous datasets.** The results presented in Figure 2 are intriguing. They demonstrate that on homophilous datasets, the homophily of the learned structures is scarcely different from the original structure, and in some cases, even lower. This is counterintuitive, considering that methods like GEN explicitly aim to increase the homophily of the learned structures. However, on heterophilous datasets in Figure 3, the homophily of the learned structures is significantly improved in most cases. This distinction may be attributed to the starting level of graph homophily: As GSL methods are trained with limited supervision signals, the number of edges that can be recovered or removed by GSL is limited. Thus, on heterophilous datasets where most edges do not fit the homophily assumption, these limited edges are more likely to be updated. On the other hand, they are already satisfied on homophilous datasets, explaining the lack of improvement here.

④ **Homophily is not always a proper guidance for structure learning.** As shown in Figure 2, Figure 3 and Table 4, homophily is only significantly positively correlated with performance on two datasets, i.e., BlogCatalog and Flickr. In most cases, we do not observe positive correlation between the performance and the homophily, in some cases even negatively correlated (on Citeseer and Wiki-cooc). These findings suggest that homophily of structure is not a proper guidance for GSL *in all cases*, which may break the common assumption and call for novel learning objective. This observation can be possibly explained by the viewpoint that certain heterophilous structural patterns can be leveraged by GNNs [29, 27], thereby guiding structure learning with homophily given limited supervision may not generate a sufficiently homophilous structure, but break these patterns and leads to suboptimal results.

Table 5: Results on Cora. GNNs are trained on structures learned from different GSL methods.

| Structure source | GCN | SGC | JKNet | APPNP | GPRGNN | LPA | LINK |
|---|---|---|---|---|---|---|---|
| Original Structure | 81.95 | 80.00 | 80.40 | 83.33 | 83.51 | 60.30 | 50.06 |
| ProGNN | 82.58 (↑0.63) | 82.10 (↑2.10) | 82.24 (↑1.84) | 83.38 (↑0.05) | 83.36 (↓0.15) | 75.85 (↑15.55) | 78.64 (↑28.58) |
| IDGL | 83.01 (↑1.06) | 83.44 (↑3.44) | 82.22 (↑1.82) | 84.34 (↑1.01) | 84.76 (↑1.25) | 77.31 (↑17.01) | 75.27 (↑25.21) |
| GRCN | 83.98 (↑2.03) | 84.66 (↑4.66) | 84.09 (↑3.69) | 83.35 (↑0.02) | 84.41 (↑0.90) | 79.03 (↑18.73) | 81.83 (↑31.77) |
| GEN | 81.74 (↓0.21) | 81.82 (↑1.82) | 81.92 (↑1.52) | 82.21 (↓1.12) | 82.16 (↓1.35) | 80.97 (↑20.67) | 79.12 (↑29.06) |
| SUBLIME | 82.69 (↑0.74) | 82.32 (↑2.32) | 81.98 (↑1.58) | 82.79 (↓0.54) | 83.59 (↑0.08) | 78.42 (↑18.12) | 81.25 (↑31.19) |

Table 6: Results on Blogcatalog. GNNs are trained on structures learned from different GSL methods.

| Structure source | GCN | SGC | JKNet | APPNP | GPRGNN | LPA | LINK |
|---|---|---|---|---|---|---|---|
| Original Structure | 76.12 | 75.37 | 74.17 | 93.72 | 93.82 | 57.53 | 64.47 |
| ProGNN | 71.73 (↓4.39) | 75.20 (↓0.17) | 73.73 (↓0.44) | 93.71 (↓0.01) | 94.70 (↑0.88) | 53.66 (↓3.87) | 66.00 (↑1.53) |
| IDGL | 89.35 (↑13.23) | 75.20 (↓0.17) | 89.77 (↑15.60) | 94.90 (↑1.18) | 95.39 (↑1.57) | 73.84 (↑16.31) | 81.25 (↑16.78) |
| GRCN | 75.69 (↓0.43) | 74.77 (↓0.60) | 75.62 (↑1.45) | 93.45 (↓0.27) | 93.53 (↓0.29) | 59.68 (↑2.15) | 66.76 (↑2.29) |
| GEN | 90.01 (↑13.89) | 89.31 (↑13.94) | 90.23 (↑16.06) | 90.37 (↓3.35) | 90.40 (↓3.42) | 86.04 (↑28.51) | 86.29 (↑21.82) |
| SUBLIME | 95.01 (↑18.89) | 94.95 (↑19.58) | 73.73 (↓0.44) | 95.35 (↑1.63) | 94.51 (↑0.69) | 89.04 (↑31.51) | 86.45 (↑21.98) |

## 4.3 Generalizability (RQ3)

We show the performance of several GNN models and simple non-GNN models on Cora and BlogCatalog using the structures learned by GSL methods as inputs in Table 5 and 6. Results on other datasets can be found in Appendix D.2. We have the following observation based on the results:

⑤ **The structures learned by GSL methods exhibit strong generalizability.** The results in Table 5 and 6 demonstrate that many GNN models show improved performance on the learned structure, marked in green, as compared to the original structure. This observation underscores the generalizability of the learned structure and its potential to enhance numerous GNN methods. Additionally, we observed that the structures learned by GSL methods also help to improve the performance of two simple non-GNN methods - LPA and LINK - and in some instances, they even outperform GNNs. The promising results strengthen the notion of the learned structure's generalizability, even without considering node features as input. In conclusion, the experimental results provide strong evidence for the generalizability of the learned structure and call for further exploration and applications of GSL methods.

## 4.4 Time and Memory Efficiency (RQ4)

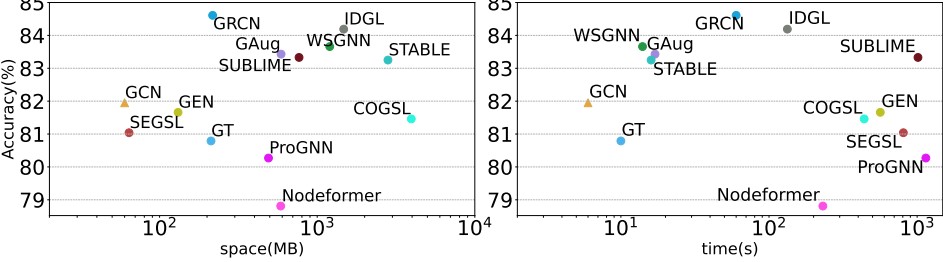

Figure 4: Time and space consumption of different methods on Cora

The efficiency of all methods on Cora are presented in Figure 4. For complete statistics on other datasets, please refer to Appendix D.3.

⑥ **Most GSL methods have large time and space consumptions.** Figure 4 clearly demonstrate that the current state-of-the-art GSL methods is struggling to achieve a satisfactory balance between performance and efficiency. In particular, most existing GSL methods suffer from significant efficiency issues, with many taking up to ten times longer to run than the GCN method. According to the results, ProGNN is the slowest, requiring 190 times longer than GCN. Likewise, most GSL methods consume an excessive amount of memory, with CoGSL consuming up to 66 times more.

The efficiency problem of GSL methods is especially pronounced on larger datasets, as discussed in Appendix D.3. According to Table 2, several GSL methods run out of memory when performing graph structure learning on the Questions dataset. Given these findings, it is vital to address the efficiency problem to ensure that GSL methods can be deployed successfully in a wide spectrum of real-world scenarios.

## 5    Future Direrctions

Drawing upon our empirical analyses, we point out some promising future directions for GSL.

**Rethinking the necessity of homophily in GSL.** While current GSL methods commonly pursue homophily within the refined graph structure, observations ③ and ④ suggest that performance improvements in GSL do not necessarily originate from increased homophily. Consequently, it is essential to rethink the necessity of homophily in GSL and explore other factors contributing to the effectiveness of GSL.

**Designing adaptive GSL methods for diverse datasets.** Observations ① and ② indicate that current GSL method do not universally work well across diverse datasets. Thus, there is an evident opportunity for creating innovative GSL methods that can adapt to diverse datasets. To achieve this goal, two critical questions arise: 1) What characteristics should learned structures exhibit for disparate datasets? Our observation ④ highlights that a sole focus on homophily may not lead to substantial improvements in certain datasets, suggesting the need to explore more reliable properties. 2) How can we incorporate these characteristics into the structure learning? Some properties might be hard to evaluate or optimize, warranting further investigation.

**Developing task-agnostic GSL methods.** Existing research efforts on GSL are mainly task-motivated. However, real-world scenarios often necessitate the refinement of a graph structure without accessing the downstream task. Although there are a few preliminary works on task-agnostic GSL [26, 22, 52], they exhibit certain limitations on preserving the naive characteristics such as proximity which is not sufficient in downstream tasks like graph clustering [12]. The core challenge is to extract semantic information from the graph data and to define optimality of the structure in the absence of explicit labels.

**Improving the efficiency of GSL methods.** Observation ⑥ exposes the issue of efficiency in GSL, requiring further attention in GSL research. Some GSL methods may run out of memory or exceed the time limit, even though the datasets we use are fairly small by today's standards [14]. The practical utility of current GSL methodologies is often hampered by these efficiency issues. Although some attempts have been made to address this problem, including limiting the weights on pre-existing edges [9] or employing anchor points [2], they commonly compromise the expressiveness of GSL. Drawing inspiration from the successful adoption of sampling strategies in Graph Neural Networks (GNNs) for acceleration [25], it would be promising to devise sophisticated sampling methodologies specifically tailored for GSL.

## 6    Conclusion and Future Work

This paper introduces a comprehensive benchmark for graph structure learning (GSL), OpenGSL, by reimplementing and comparing thirteen cutting-edge methods across a diverse set of datasets. The fair comparison and comprehensive analysis unearth several key findings on this promising research topic. Firstly, we observe that GSL methods do not consistently outperform vanilla GNNs across diverse datasets. Secondly, we revisit the correlation between homophily and structure learning, encouraging innovative learning objectives other than homophily. Thirdly, we empirically demonstrate the generalizability of the learned structure. Lastly, we highlight the efficiency problem of the existing methods, emphasizing the need to develop scalable GSL methods. We believe that this benchmark will have a positive impact on this emerging research domain. We have made our code publicly available and welcome further contributions of new datasets and methods.

We plan to further enhance OpenGSL in the following ways. To begin with, we plan to broaden our dataset coverage to ensure a more comprehensive evaluation. Specifically, we will look into heterogeneous networks that are used widely in practical applications. In view of the efficiency limitations in current GSL methods, we will explore ways to apply structure learning on large-scale datasets such as OGB [14]. Lastly, in this study we only focus on node classification, which is

widely adopted in existing GSL research. As there have been attempts to extend GSL to other tasks (e.g., graph classification [39], clustering [12]), we plan to expand the support of OpenGSL to other graph learning tasks for a more comprehensive coverage. We will update our repository to reflect all the potential new tasks, datasets, as well as any improvement or correction. We are also open to suggestions and welcome any comments to improve our benchmark and elevate its usability.

## Acknowledge

This work is supported by the Starry Night Science Fund of Zhejiang University Shanghai Institute for Advanced Study, China (Grant No: SN- ZJU-SIAS-001), the National Natural Science Foundation of China (621062219, 62372399), Zhejiang Provincial Natural Science Foundation of China (Grant No: LTGG23F030005), Ningbo Natural Science Foundation (Grant No: 2022J183) and the advanced computing resources provided by the Supercomputing Center of Hangzhou City University.

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

# A   Additional Details on Benchmark

## A.1   Datasets

Table 7: Statistics of the datasets used in the benchmark. $\overline{CC_l}$ denotes average local clustering coefficient. $CC_g$ denotes global clustering coefficient. $h_{edge}$ denotes edge homophily. $h_{adj}$ and "LI" denotes adjusted homophily and label informativeness proposed in [33], respectively.

| | Nodes | Edges | Feat | Avg degree | Classes | $\overline{CC_l}$ | $CC_g$ | $h_{edge}$ | $h_{adj}$ | LI |
|---|---|---|---|---|---|---|---|---|---|---|
| **Cora** | 2,708 | 5,278 | 1,433 | 3.9 | 7 | 0.24 | 0.09 | 0.81 | 0.77 | 0.59 |
| **Citeseer** | 3,327 | 4,552 | 3,703 | 2.7 | 6 | 0.14 | 0.13 | 0.74 | 0.67 | 0.45 |
| **Pubmed** | 19,717 | 44,324 | 500 | 4.5 | 3 | 0.06 | 0.05 | 0.80 | 0.69 | 0.41 |
| **Questions** | 48,921 | 153,540 | 301 | 6.3 | 2 | 0.06 | 0.02 | 0.84 | 0.02 | 0.00 |
| **Minesweeper** | 10,000 | 39,402 | 7 | 7.9 | 2 | 0.44 | 0.43 | 0.68 | -0.05 | 0.00 |
| **BlogCatalog** | 5,196 | 171,743 | 8,189 | 66.1 | 6 | 0.12 | 0.08 | 0.40 | 0.27 | 0.09 |
| **Flickr** | 7,575 | 239,738 | 12,047 | 63.3 | 9 | 0.33 | 0.10 | 0.24 | 0.14 | 0.03 |
| **Amazon-ratings** | 24,492 | 93,050 | 300 | 7.6 | 5 | 0.58 | 0.32 | 0.38 | 0.14 | 0.04 |
| **Roman-empire** | 22,662 | 32,927 | 300 | 2.9 | 18 | 0.39 | 0.29 | 0.05 | -0.05 | 0.11 |
| **Wiki-cooc** | 10,000 | 2,243,042 | 100 | 448.6 | 5 | 0.55 | 0.23 | 0.34 | -0.03 | 0.02 |

**Cora, Citeseer and Pubmed** [36] are three citation networks commonly used in prior GSL works [2, 16, 53, 42, 43, 26], with nodes representing papers and edges representing papers' citation relationships. Node features are bag-of-words feature vectors. The label of each node is its category of research topic.

**BlogCatalog** [15] is a social network created from an online community where bloggers can follow each other. The features associated with each user are generated based on the keywords in their blog description, while the labels are chosen from a set of predefined categories based on interests of bloggers.

**Flickr** [15] is a platform for sharing images and videos where users can follow each other, forming a social network. In this dataset, user-specified interest tags are used to generate features, with the groups they have joined serving as labels.

Five datasets from [34] are listed below.

**Amazon-ratings** is based on the Amazon product co-purchasing network dataset [21]. In this dataset, nodes represent products while edges connect products that are frequently bought together. Node features are the mean of FastText embeddings for words present in the product description. The ratings of products are divided into five distinct classes as labels.

**Roman-empire** is based on the Roman Empire article from English Wikipedia. In this dataset, each node corresponds to a single word in the text. When two words are consecutive in the text or are connected by a dependency tree, an edge is created between them. Node features are FastText word embeddings. The label of a node is its syntactic role.

**Questions** is based on a question-answering website. In this dataset nodes represent users, and two users are connected if one user answered the other user's question during a given time interval. Node features are the mean of FastText embeddings for words in the user's description. The label of a node is whether the user remained active on the website. Notably, the dataset is *highly imbalanced* with 97% of users belonging to the active class.

**Minesweeper** is a synthetic dataset based on the Minesweeper game. Each node represents a square in a $100 \times 100$ grid. Two nodes are connected if they are adjacent in the grid. 20% of the nodes are randomly designated as mines for prediction. Node features are one-hot-encoded numbers of neighboring mines, while for 50% of the selected nodes the features are set unknown to increase task difficulty, indicated by a separate binary feature. Notably, the dataset is also *highly imbalanced*.

**Wiki-cooc** is based on the English Wikipedia. In this dataset, nodes represent unique words and edges connect frequently co-occurring words. Node features are FastText word embeddings. The label of a node is its part of speech.

## A.2 Algorithms

Here we introduce all the GSL algorithms implemented in OpenGSL.

**LDS** [10] assumes the structure is sampled from mutually independent Bernoulli distributions with a total of $n^2$ parameters. This paper proposes a bilevel optimization problem, where the inner problem optimizes GNN parameters on the training set, while the outer problem optimizes the structure parameters on the validation set. A meta-learning-based approach is adopted to optimize the structure parameters.

**ProGNN** [16] optimizes the adjacency matrix directly, which is set as an $n \times n$ parameter matrix. Three properties, namely sparsity, low-rankness, and smoothness, are introduced to guide structure learning.

**IDGL** [2] models the structure as a weighted cosine function of node representations. To increase efficiency for larger graphs, the paper proposes an anchor-based structure learning method.

**GRCN** [49] incorporates two GNNs, one for node classification and the other one for computing node representations that are used to derive structure via a metric function. Both GNNs are optimized simultaneously to minimize the task loss.

**GAug** [53] shares a similar architecture with GRCN, differing in that GAug employs a graph auto-encoder to learn the structure, while structure learning is guided by an edge-prediction loss in addition to the task loss.

**SLAPS** [9] explores the scenario where original structure is absent. It leverages a MLP to obtain node representations to generate structure through a metric function. The overall architecture of SLAPS is similar to GRCN, with an additional auto-denoising loss to guide structure learning. Specifically, the corrupted features and the learned structure are fed into another GNN, and the output is expected to reconstruct the original features.

**GEN** [42] assumes that the optimal structure is generated by an SBM model, and further assumes that the similarity matrices of the node representations at different levels are observations of the optimal structure. The EM algorithm is employed to learn the expected optimal structure given a well-optimized GNN. GEN performs structural learning and task learning in an iterative way.

**Nodeformer** [43] is a model that allows for layer-wise edge reweighting on all node pairs by utilizing a kernelized Gumbel-Softmax operator, which reduces complexity from quadratic to linear with respect to the number of nodes. To guide structure learning, an extra edge-level regularization is implemented.

**CoGSL** [24] extracts two fundamental views from the original graph and refines them using a view estimator. An adaptive fusion strategy is employed to obtain the final view. CoGSL maintains the performance of three views while reducing the mutual information between every two views to achieve a "minimal sufficient structure."

**SUBLIME** [26] presents an approach for unsupervised structure learning. It proposes a structure bootstrapping contrastive learning framework, where an anchor structure is set up to provide supervision signals for the learner structure. Specifically, SUBLIME utilizes a GNN-based encoder to obtain node representations from both views and optimizes the GNN encoder through a node-level contrastive loss. During the training process, the anchor structure is updated every several epochs as the interpolation between the anchor structure and the learner structure.

**WSGNN** [20] is a probabilistic generative model that utilizes variational inference to jointly learn node labels and graph structure. It employs a two-branch model architecture to collectively refine node embeddings and latent structure. A composite loss function is derived from the underlying data distribution, effectively capturing the interplay between observed data and missing data.

**STABLE** [22] obtains reliable node representations leveraging contrastive learning. The new structure is computed as the similarity matrix of node representations. In addition, an advanced GCN is proposed to enhance the robustness of the ordinary GCN.

**SEGSL** [59] introduces the concept of graph structural entropy. It starts by enhancing the structure guided by the one-dimensional structural entropy maximization strategy. Then, an encoding tree is constructed to capture the hierarchical information of the graph structure. Finally, SEGSL reconstructs

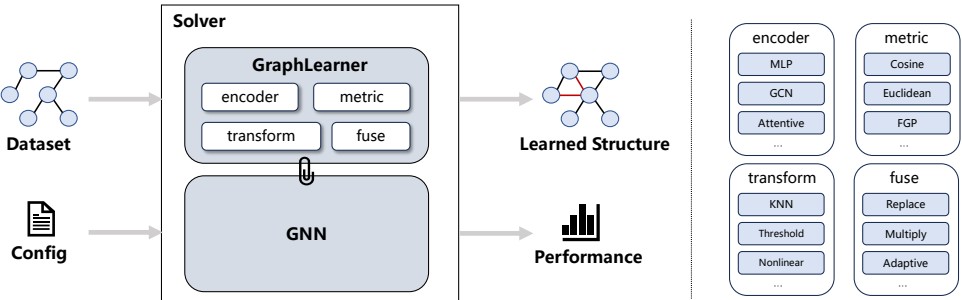

Figure 5: Code structure of OpenGSL. For a specified dataset and configuration, a solver will return the learned structure and performance on the task. `Solver` serves as a foundational class that supports different implemented methods. It includes two main modules, `GraphLearner` and `GNN`, and controls the training and testing processes. `GraphLearner` consists of four components, each of which can be easily replaced (as shown on the right).

the graph structure from the encoding tree. The method performs structure learning and task learning in an iterative way.

### A.3 Discussion on Related GNN Benchmarks

There have been considerable efforts benchmarking GNNs [37, 14, 8, 23]. To name a few, Open Graph Benchmark (OGB) [14] released a diverse set of large-scale and challenging datasets covering various graph machine learning tasks. [8] introduces a collection of medium-sized datasets to quantify progress and support GNN research.

Our work differs from existing benchmarks in several aspects. Firstly, to the best of our knowledge, we are the first to propose a benchmark tailored for Graph Structure Learning (GSL). We have made significant efforts to carefully reimplement a wide range of cutting-edge GSL algorithms. OpenGSL provides a fair evaluation of these algorithms, and a user-friendly platform for conducting experiments, which is not attainable with existing benchmarks. Secondly, we consider both homophilous and heterophilious datasets when evaluating GSL—a dimension that has been largely overlooked in existing benchmarks for GNNs. Lastly, in addition to node classification(Section 4.1), we include some experimental settings that are specific to GSL, such as experiments on homophily/heterophily(Section 4.2), generalizability of learned structures(Section 4.3), and results in the absence of original structures(Appendix D.7). These experimental settings allow us to delve deeper into GSL, based on which we have provided valuable insights and promising research directions. We believe that OpenGSL, as a benchmark targeting GSL research, can have a positive impact on this emerging research field.

## B  Package

We have developed a open-sourced package Open Graph Structure Learning (OpenGSL)[4], which offers a comprehensive and unbiased platform for evaluating GSL algorithms and facilitating future research in this domain.

As shown in Figure 5, the code structure is well organized to ensure fair experimental settings across algorithms, easy reproduction of the experimental results and convenient trials on flexibly assembled models. OpenGSL consists of the following key modules. The `Config` module includes the files that define the necessary hyperparameters and settings. The `Dataset` module is used to load datasets. The `Solver` consisting of `GraphLearner` and `GNN` controls the process of training and evaluation, which serves as the base class for various reproduced methods. `GraphLearner` is composed of four elements, each of which supports multiple design choices. Given

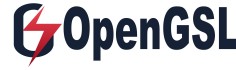

Figure 6: OpenGSL

---

[4]https://github.com/OpenGSL/OpenGSL

Table 8: Hyper-parameter search space of all implemented methods.

| Algorithm | Hyper-parameter | Search Space |
|---|---|---|
| General Settings | learning rate | 1e-1, 1e-2, 1e-3, 1e-4 |
| | weight decay | 5e-4, 5e-5, 5e-6, 5e-7, 0 |
| GCN [18] | number of layers | 2, 3, 4, 5 |
| | hidden size | 16, 32, 64, 128 |
| | dropout | 0, 0.2, 0.5, 0.8 |
| GRCN [49] | $K$ for nearest neighbors | 5, 50, 100, 200 |
| | learning rate for graph | 1e-1, 1e-2, 1e-3, 1e-4 |
| SLAPS [9] | learning rate of DAE | 1e-2, 1e-3 |
| | dropout_adj1 | 0.25, 0.5 |
| | dropout_adj2 | 0.25, 0.5 |
| | $k$ for nearest neighbors | 10, 15, 20, 30 |
| | $\lambda$ | 0.1, 1, 10, 100, 500 |
| | ratio | 1,5,10 |
| | nr | 1, 5 |
| WSGNN [20] | hidden size | 16, 32, 64, 128 |
| | graph skip connection $\beta$ for q model | 0.5, 0.6, 0.7, 0.8, 0.9 |
| | number of heads in graph learner | 1, 2, 4, 6, 8 |
| CoGSL [24] | ve_lr | 1e-1, 1e-2, 1e-3, 1e-4 |
| | ve_dropout | 0, 0.2, 0.5, 0.8 |
| | $\tau$ | 0, 0.2, 0.5, 0.8 |
| | $\epsilon$ | 0.1, 1 |
| | $\lambda$ | 0, 0.2, 0.5, 0.8, 1 |
| Nodeformer [43] | number of layers | 2, 3, 4, 5 |
| | hidden size | 16, 32, 64, 128 |
| | number of heads | 1, 2, 4, 8 |
| | dropout | 0, 0.2, 0.5, 0.8 |
| | number of samples for gumbel softmax sampling | 5, 10, 20 |
| | weight for edge reg loss | 1, 0.1, 0.01 |
| GEN [42] | $k$ for nearest neighbors | 7, 8, 9, 10 |
| | tolerance for EM | 1e-2, 1e-3, 1e-4 |
| | threshold for adding edges | 0.5, 0.6, 0.7, 0.8 |
| SEGSL [59] | $K$ | 2, 3, 4 |
| | $se$ | 2, 3, 4 |
| LDS [10] | $\tau$ | 5, 10, 15 |
| ProGNN [16] | learning rate for adj | 1e-1, 1e-2, 1e-3, 1e-4 |
| GAug [51] | $\alpha$ for interpolation | 0, 0, 0.1, 0.3, 0.5, 0.7, 0.9, 1 |
| | temperature | 0.3, 0.6, 0.9, 1.2 |
| | epochs for warm up | 0, 10, 20 |
| IDGL [2] | number of anchors | 300, 500, 700 |
| | number of heads in structure learning | 2, 4, 6, 8 |
| | $\lambda_1$ for interpolation | 0.7, 0.8, 0.9 |
| | $\lambda_2$ for interpolation | 0.1, 0.2, 0.3 |
| SUBLIME [26] | dropout rate for edge | 0, 0.25, 0.5 |
| | $\tau$ for bootstrapping | 0.99, 0.999, 0.9999 |
| STABLE [22] | threshold for consine similarity | 0.1, 0.2, 0.3 |
| | $k$ for nearest neighbors | 1, 3, 5, 7 |

specific dataset and config file, a solver will return the learned structure and the task performance. For more details and updated features, please refer to our Github repository.

# C Additional Details on Experimental Settings

**Running Experiments.** Our experiments are mostly conducted on a Linux server with an Intel(R) Xeon(R) Silver 4216 CPU @ 2.10GHz, 125 GB RAM, and an NVIDIA GTX 2080 Ti GPU (12GB). However, as some methods may exceed GPU memory, we run partial experiments (including all

experiments related to efficiency) on another Linux server with an Intel(R) Xeon(R) Platinum 8358 CPU @ 2.60GHz, 1008 GB RAM, and an NVIDIA A800 GPU (80GB).

**General Experimental Settings.** We strive to adhere to the original implementation of various GSL methods provided in their respective papers or source codes. To achieve this, we have integrated different options into a standardized framework as shown in Figure. We choose not to augment the models with LayerNorm as in [34], since this technique has not been adopted by GSL algorithms. Consequently, there may be some discrepancies observed in the results for the five datasets from [34].

**Hyperparamter.** We conduct comprehensive hyperparameter tuning to ensure a thorough and impartial evaluation of these GSL methods. When the paper and source code of a particular method do not provide information on hyperparameter settings, we tune the corresponding hyperparameters. Hyperparameter tuning is conducted via bayesian search using NNI[5]. The hyperparameter search spaces of all methods are presented in Table 8. For detailed meanings of these hyperparameters please refer to their original papers. More complete hyperparameter settings for all the methods can be found in our Github repository [6].

# D  Additional Results

## D.1  Additional Results on the Properties of Learned Structure

In Section 4.2, we analyze the correlation between homophily and performance. The results show that homophily is not always positively correlated with performance and therefore not a suitable guidance in all scenarios. Here, we present additional results on two new measures named "adjusted homophily" and "label informativeness", introduced by [33].

Adjusted homophily is proposed to overcome the sensitivity of commonly used homophily measures to the number of classes and their balance. Specifically, the adjusted homophily of a graph $\mathcal{G}$ is defined as follows:

$$\text{adj\_homo}(\mathcal{G}) = \frac{\text{homo}(\mathcal{G}) - \sum_{k=1}^{C} p(k)^2}{1 - \sum_{k=1}^{C} p(k)^2} \tag{2}$$

where $p(k) = \frac{\sum_{v_i : y_i = k} d(v_i)}{2|E|}$, $d(v_i)$ is the degree of node $v_i$.

Label informativeness measures the overall informativeness of neighbor's labels for central node's labels in a graph. This metric, which goes beyond the homophily-heterophily dichotomy, can further describe the degree of regularity in the connectivity patterns of heterophilous graphs. The label informativeness of a graph $\mathcal{G}$ is defined as follows:

$$\text{LI}(\mathcal{G}) = 2 - \frac{\sum_{c_1, c_2} p(c_1, c_2) \log p(c_1.c_2)}{\sum_c p(c) \log p(c)} \tag{3}$$

where $p(c_1, c_2) = \frac{|(u,v)|(u,v) \in \mathcal{E}, y_u = c_1, y_v = c_2|}{2|\mathcal{E}|}$.

The adjusted homophily and label informativeness of structures learned by GSL methods on various datasets are presented in Figure 7 and Figure 8, respectively.

The figures reveal a similarity in the patterns of adjusted homophily, label informativeness and homophily for structures learned by GSL methods. Notably, adjusted homophily and label informativeness show significant positive correlations with performance only on BlogCatalog and Flickr, which is the same as homophily. These results imply that the newly proposed measures are still unable to effectively gauge structure quality and unsuitable for guiding structure learning on real data, despite that they are found to be effective on synthetic data [33]. Further exploration is needed in this aspect.

---

[5]https://github.com/microsoft/nni/
[6]https://github.com/OpenGSL/OpenGSL/tree/main/opengsl/config

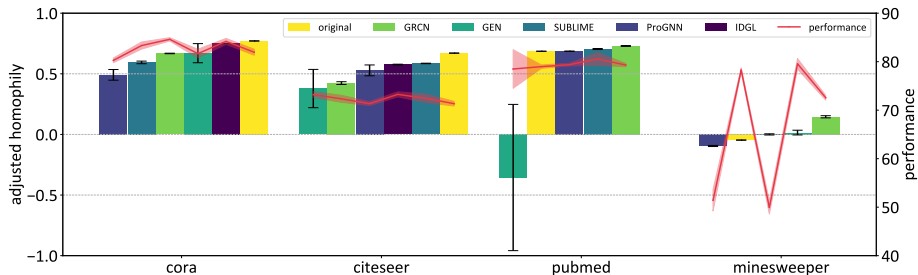

Figure 7: Adjusted homophily of learned structures and performances on homophilous datasets. The methods are ordered by homophily. Left to right on Minesweeper: ProGNN, original, SUBLIME, GEN, GRCN.

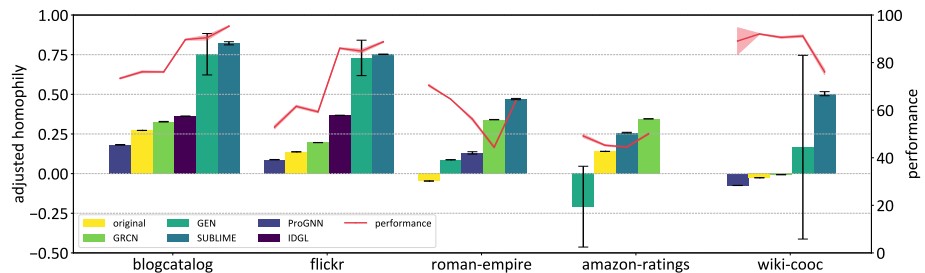

Figure 8: Adjusted homophily of learned structures and performances on heterophilous datasets. The methods are ordered by homophily. Left to right on Wiki-cooc: ProGNN, original, GRCN, GEN, SUBLIME.

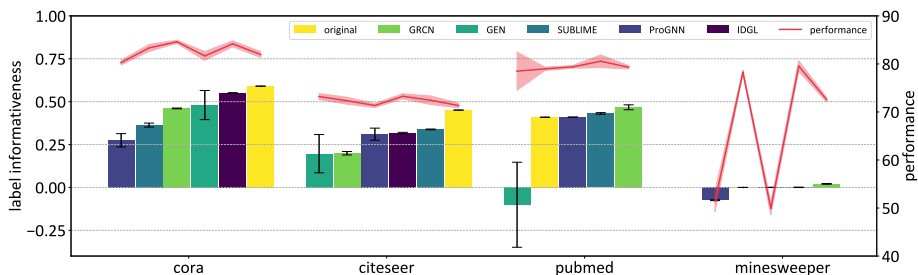

Figure 9: Label informativeness of learned structures and performances on homophilous datasets. The methods are ordered by homophily. Left to right on Minesweeper: ProGNN, original, SUBLIME, GEN, GRCN.

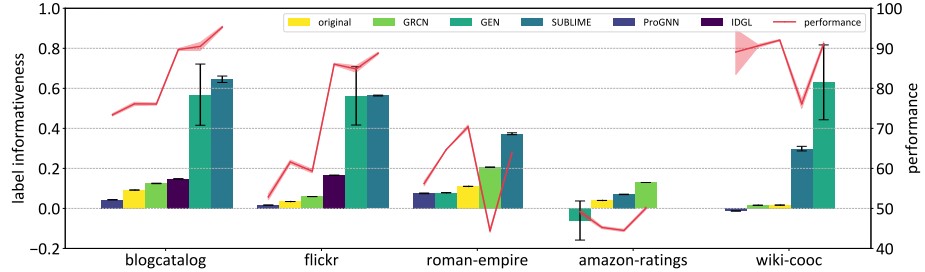

Figure 10: Label informativeness of learned structures and performances on heterophilous datasets. The methods are ordered by homophily.

## D.2 Additional Results on Generalizability

In addition to Section 4.3, we also assessed the generalizability of structures learned by various GSL methods on other datasets where GSL methods have made progress. The results are presented in Tables 9 - 11.

These additional results are consistent with the conclusions drawn in Section 4.3. In most cases, the learned structures exhibit strong generalizability. These learned structures frequently outperform the original structures and impart additional advantages to various GNN models.

Table 9: Generalizability on Citeseer. Improvements over the original structure are marked green.

| Structure source | GCN | SGC | JKNet | APPNP | GPRGNN | LPA | LINK |
|---|---|---|---|---|---|---|---|
| Original Structure | 71.34 | 71.20 | 68.26 | 71.86 | 71.14 | 26.20 | 34.70 |
| ProGNN | 70.00 (↓1.34) | 69.02 (↓2.18) | 68.33 (↑0.07) | 71.14 (↓0.72) | 70.45 (↓0.69) | 64.46 (↑38.26) | 57.12 (↑22.42) |
| IDGL | 73.03 (↑1.69) | 73.20 (↑2.00) | 72.21 (↑3.95) | 73.82 (↑1.96) | 72.96 (↑1.82) | 64.17 (↑37.97) | 64.87 (↑30.17) |
| GRCN | 70.90 (↓0.44) | 71.44 (↑0.24) | 71.03 (↑2.77) | 73.33 (↑1.47) | 73.25 (↑2.11) | 70.43 (↑44.23) | 69.10 (↑34.40) |
| GEN | 72.83 (↑1.49) | 73.02 (↑1.82) | 72.67 (↑4.41) | 73.24 (↑1.38) | 73.27 (↑2.13) | 65.55 (↑39.35) | 68.20 (↑33.50) |
| SUBLIME | 71.22 (↓0.12) | 70.76 (↓0.44) | 68.17 (↓0.09) | 72.60 (↑0.74) | 71.64 (↑0.50) | 53.50 (↑27.30) | 55.00 (↑20.30) |

Table 10: Generalizability on Pubmed. Improvements over the original structure are marked green.

| Structure source | GCN | SGC | JKNet | APPNP | GPRGNN | LPA | LINK |
|---|---|---|---|---|---|---|---|
| Original Structure | 78.98 | 78.96 | 78.00 | 80.10 | 79.05 | 24.70 | 44.71 |
| GEN | 79.50 (↑0.52) | 79.23 (↑0.27) | 79.14 (↑1.14) | 79.83 (↓0.27) | 79.76 (↑0.71) | 51.61 (↑26.91) | 64.79 (↑20.08) |
| ProGNN | 78.74 (↓0.24) | 78.70 (↓0.26) | 76.97 (↓1.03) | 80.33 (↑0.23) | 79.10 (↑0.05) | 24.80 (↑0.10) | 44.23 (↓0.48) |
| GRCN | 79.15 (↑0.17) | 79.12 (↑0.16) | 77.92 (↓0.08) | 80.28 (↑0.18) | 79.41 (↑0.36) | 44.82 (↑20.12) | 44.57 (↓0.14) |
| SUBLIME | 79.24 (↑0.26) | 79.67 (↑0.71) | 78.90 (↑0.90) | 80.99 (↑0.89) | 80.49 (↑1.44) | 30.60 (↑5.90) | 49.58 (↑4.87) |

Table 11: Generalizability on Flickr. Improvements over the original structure are marked green.

| Structure source | GCN | SGC | JKNet | APPNP | GPRGNN | LPA | LINK |
|---|---|---|---|---|---|---|---|
| Original Structure | 61.60 | 60.72 | 57.69 | 83.58 | 82.31 | 39.22 | 42.60 |
| ProGNN | 47.28 (↓14.32) | 60.79 (↑0.07) | 61.63 (↑3.94) | 84.16 (↑0.58) | 86.14 (↑3.83) | 40.14 (↑0.92) | 50.15 (↑7.55) |
| GRCN | 61.14 (↓0.46) | 58.86 (↓1.86) | 60.52 (↑2.83) | 83.14 (↓0.44) | 82.11 (↓0.20) | 44.95 (↑5.73) | 50.79 (↑8.19) |
| IDGL | 85.98 (↑24.38) | 86.43 (↑25.71) | 84.97 (↑27.28) | 87.98 (↑4.40) | 88.75 (↑6.44) | 71.18 (↑31.96) | 80.69 (↑38.09) |
| GEN | 85.94 (↑24.34) | 85.93 (↑25.21) | 85.86 (↑28.17) | 84.16 (↑0.58) | 85.92 (↑3.61) | 84.15 (↑44.93) | 84.47 (↑41.87) |
| SUBLIME | 88.65 (↑27.05) | 88.20 (↑27.48) | 88.00 (↑30.31) | 89.55 (↑5.97) | 89.55 (↑7.24) | 77.74 (↑38.52) | 71.86 (↑29.26) |

## D.3 Addtional Results on Efficiency

We record the efficiency of existing GSL methods on more datasets, and the results are displayed in Figures 11- 15.

Based on these results, it is evident that the current GSL methods faces challenges in efficiency, and the problem is more severe on large-scale graphs. Henceforth, there is a pressing demand for the development of more efficient GSL methods capable of processing large-scale graph data.

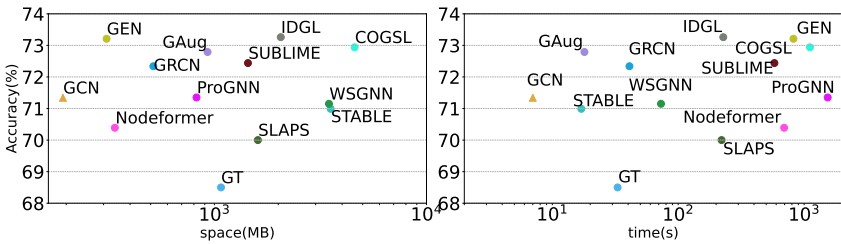

Figure 11: Time and space consumption of different methods on Citeseer

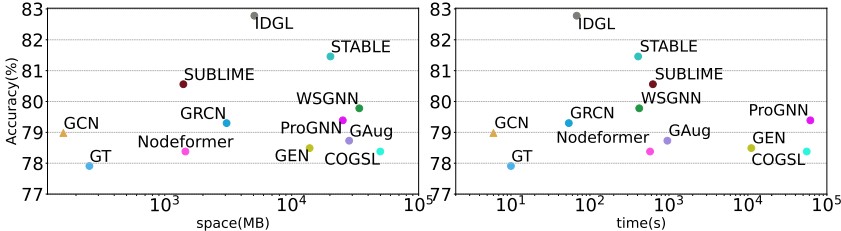

Figure 12: Time and space consumption of different methods on Pubmed

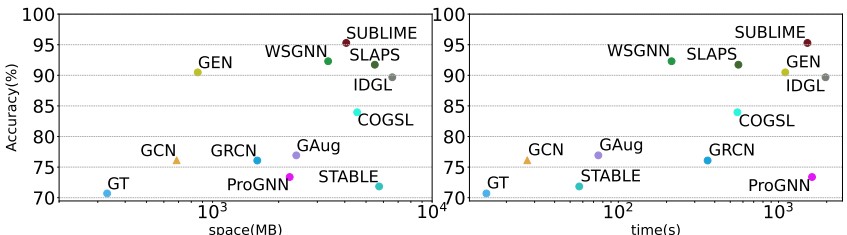

Figure 13: Time and space consumption of different methods on BlogCatalog

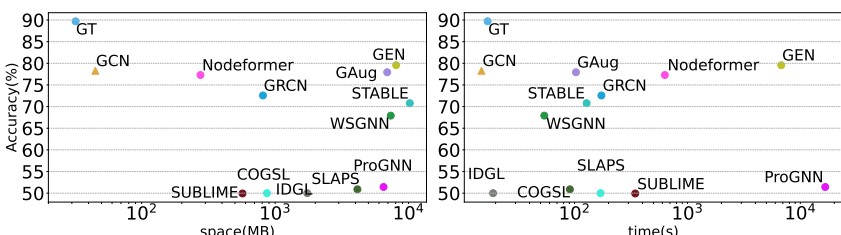

Figure 14: Time and space consumption of different methods on Minesweeper

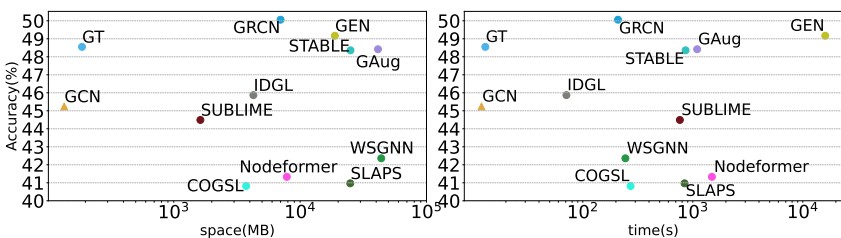

Figure 15: Time and space consumption of different methods on Amazon-ratings

## D.4    Addtional Results on Robustness

GSL is capable of refining the original suboptimal graph structure, which theoretically gives it superior robustness compared to GCN. We conduct experiments to evaluate the robustness of some GSL methods. Following [56, 22], we consider two types of attacks: **Metattack** and **Random**. In Metattack, we use GCN as the surrogate model and apply Metattack [60] to perturb the graph structure to varying degrees. In Random, we randomly add noisy edges to the graph structure to introduce varying degrees of homophily. The performances of various GSL methods under these two attacks are shown below. Figure 16-17 show that most GSL methods have a slower rate of decline than GCN, indicating their better robustness compared with GCN. Among them, SUBLIME demonstrates excellent robustness when facing different attacks.

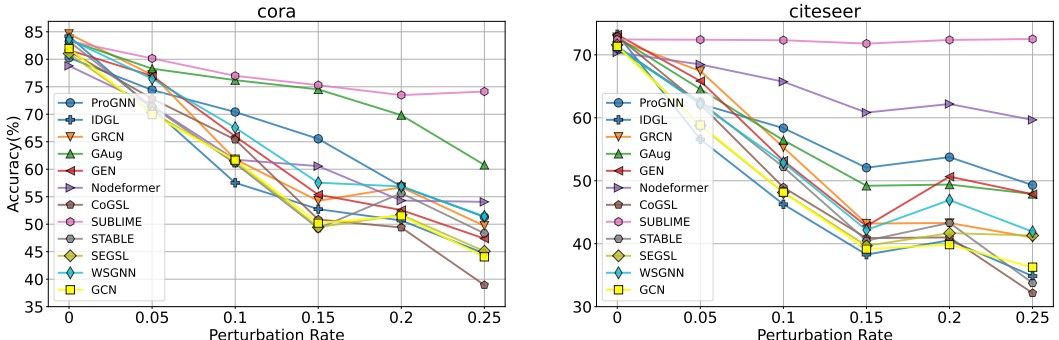

Figure 16: Robustness of GSL methods on Cora and Citeseer under Metattack.

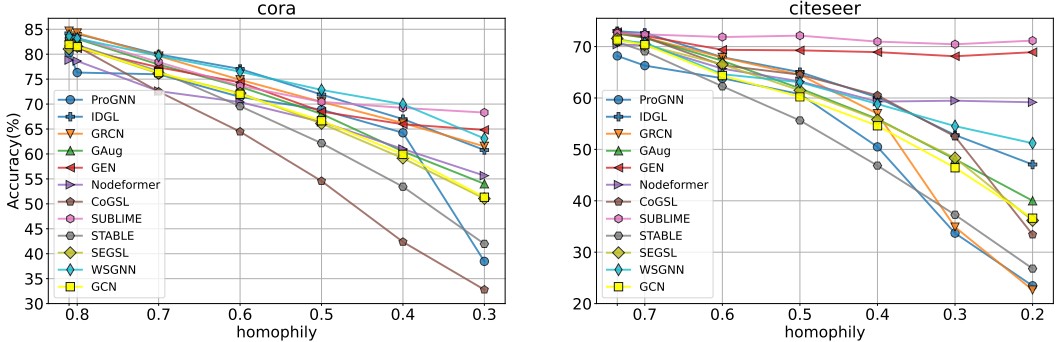

Figure 17: Robustness of GSL methods on Cora and Citeseer under Random attack.

## D.5    Additional Results on Different Backbones

The results presented in previous parts are all obtained from GSL methods using GCN as the backbone. Here, we consider two additional backbones, namely APPNP and GIN, and evaluate the performance of GSL methods with these backbones. Figures 18-20 show the experimental results on Cora, BlogCatalog, and Roman-empire.

The figures clearly illustrate that, when compared to vanilla GNN backbones, most GSL methods exhibit improvements on Cora and BlogCatalog, while do not work on Roman-empire regardless of the backbone used. This is consistent with the observations in Section 4.1. Moreover, when comparing the performance across different backbones, it is evident that GSL's performance can be further enhanced with more advanced backbones. These results indicate that GSL and GNN backbone, which are two parallel research fields, can benefit from each other.

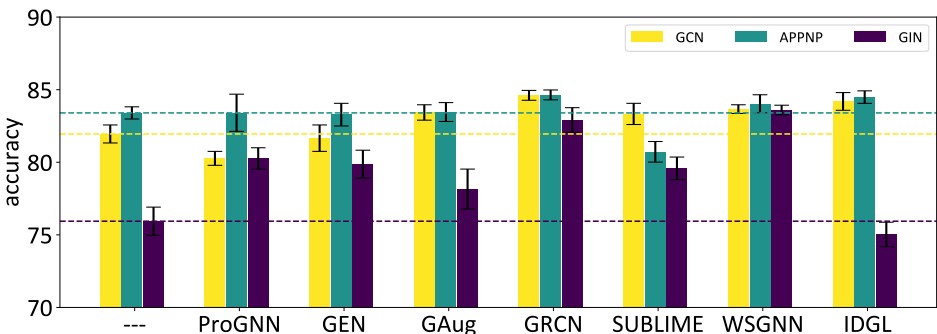

Figure 18: GSL methods with different backbones on Cora. '—' in the left stands for the vanilla backbones.

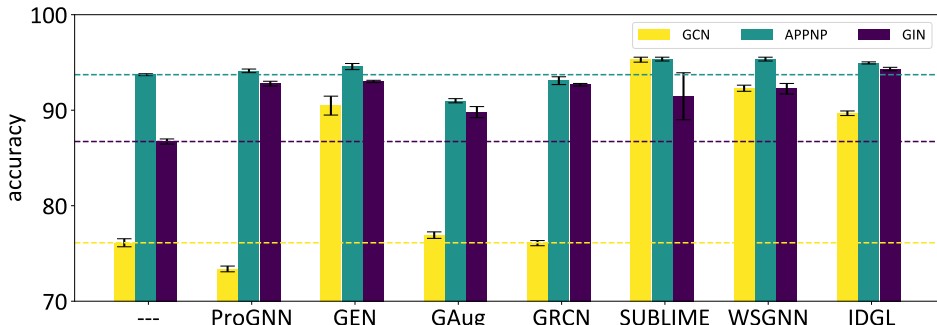

Figure 19: GSL methods with different backbones on BlogCatalog. '—' in the left stands for the vanilla backbones.

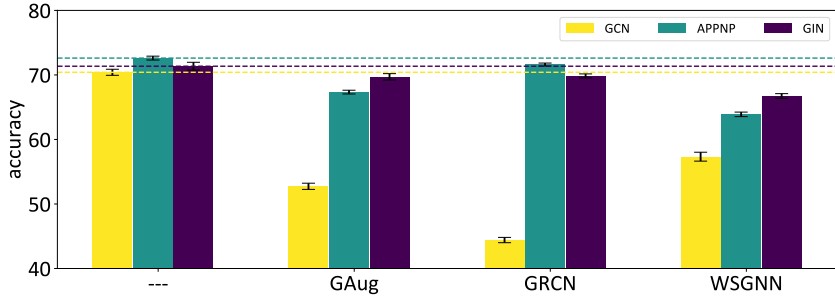

Figure 20: GSL methods with different backbones on Roman-empire. '—' in the left stands for the vanilla backbones. Some methods are not included due to OOM or exceeding time limit.

One clarification we need to make is the distinction between this experiment and the one conducted in Section 4.3. In both cases, we combine GSL with various GNNs. However, the difference lies in that the performances here are obtained from GNNs that are jointly optimized with structure, while in the other setting, we treat each GSL method as a pre-processing step and train a GNN model from scratch using the learned structure. The purpose of the former experiment is to investigate the impact of the choices on backbones, while the latter aims to examine whether the improvement of GSL stems from learning superior structures.

## D.6 Additional Results on Synthetic Graphs

In addition to the experiments conducted on real-world datasets, it is essential to include synthetic datasets as well. Synthetic datasets offer the advantage of enabling precise control over the homophily of the generated graph, facilitating more fine-grained empirical studies. Here, we adopt the CSBM model to construct synthetic datasets [7]. Following [3], we vary the connectivity probabilities to generate graph data with different structural patterns, while ensuring that the generated data satisfies the information-theoretic threshold. In the generated dataset, structures with high homophily and low homophily are both informative, which is different from the setting described in Appx. C.4.

Figure 21 shows the performance of GSL methods on synthetic datasets with different levels of homophily. We can observe that most methods perform well under high homophily conditions but poorly under low homophily conditions. However, some methods, such as SUBLIME and GEN, perform well even under low homophily. We can also see that the performance of some methods is better at 0.2 than at 0.4. These observations indicate that these methods capture informative heterophilious structures. Further exploration is needed to address real-world data, which is relatively more complex.

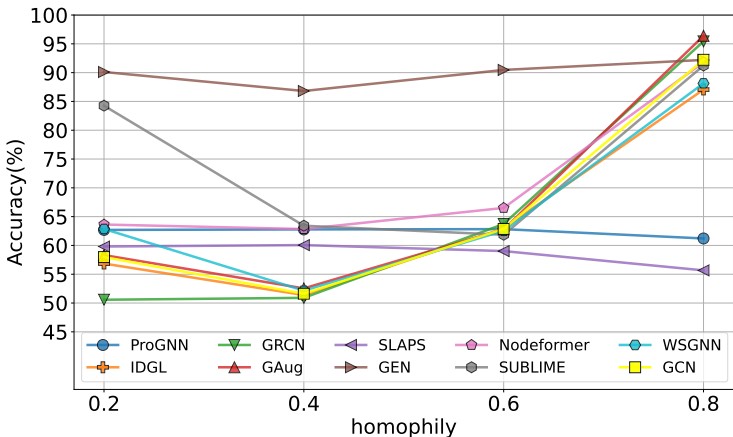

Figure 21: GSL methods om synthetic data with varying homophily.

## D.7 Additional Results without Original Structures

A promising direction for GSL is to apply GNN models to data without graph structure, thereby enabling the use of GNNs for a wider range of machine learning tasks. Traditional approaches involve using GNN on a pre-defined KNN graph. Intuitively, GSL methods should better capture the relationships between samples through joint optimization of the structure and GNN model. In this context, we conduct some preliminary experiments. Specifically, we replace the original structure of graph datasets with an identity matrix or the KNN matrix computed from node features, in which case the models have zero knowledge on the graph structures. The results are presented in Table 12.

It is evident that most GSL methods exhibit improvements compared with GCN on Cora and BlogCatalog, demonstrating the potential of GSL to extend to non-graph data. However, few methods surpass GCN on Roman-empire. Further exploration is required to gain a deeper understanding in this regard.

Table 12: Performance of GSL methods without knowledge on the structure."I" denotes the use of the identity matrix, while "K" denotes the use of the feature-based KNN matrix. "−" denotes out of memory or time limit of 24 hours exceeded.

| Model | Cora(I) | Cora(K) | BlogCatalog(I) | BlogCatalog(K) | Roman-empire(I) | Roman-empire(K) |
|---|---|---|---|---|---|---|
| GCN | 59.00 ± 0.68 | 54.79 ± 0.85 | 85.09 ± 0.51 | 81.86 ± 0.22 | 65.98 ± 0.29 | 65.29 ± 0.36 |
| LDS | 68.53 ± 0.63 | 65.56 ± 1.33 | 88.26± 0.49 | 86.72 ± 0.31 | − | − |
| ProGNN | 53.06 ± 2.98 | 53.50 ± 2.63 | 85.45 ± 0.63 | 81.50 ± 0.25 | − | − |
| IDGL | 66.54 ± 1.13 | 61.38 ± 0.53 | 91.21 ± 0.29 | 87.85 ± 0.29 | 64.85 ± 0.35 | 55.17 ± 9.54 |
| GRCN | 70.23 ± 0.43 | 62.03 ± 1.02 | 89.68 ± 0.48 | 82.79 ± 0.47 | 64.86 ± 0.28 | 63.86 ± 0.29 |
| GAug | − | 58.98 ± 0.84 | − | 81.92 ± 0.40 | − | 61.17 ± 0.51 |
| SLAPS | 72.29 ± 1.01 | 72.29 ± 1.01 | 91.73 ± 0.40 | 91.73 ± 0.40 | 65.35 ± 0.45 | 65.35 ± 0.45 |
| GEN | 64.60 ± 0.95 | 63.87 ± 1.03 | 88.50 ± 0.63 | 85.15 ± 0.42 | − | − |
| WSGNN | 67.27 ± 1.22 | 64.73 ± 1.03 | 90.30 ± 0.37 | 88.87 ± 0.27 | 65.78 ± 0.19 | 65.19 ± 0.30 |
| Nodeformer | 51.38 ± 1.56 | 57.73 ± 1.72 | 53.43 ± 11.32 | 39.60 ± 12.26 | 51.18 ± 3.12 | 53.76 ± 1.89 |
| CoGSL | 59.68 ± 1.44 | 61.93 ± 0.92 | 83.68 ± 0.19 | 85.07 ± 0.45 | 63.94 ± 0.14 | 63.49 ± 0.10 |
| SUBLIME | 69.72 ± 0.97 | 69.06 ± 0.97 | 90.91 ± 0.40 | 82.78 ± 0.76 | 64.77 ± 0.41 | 64.38 ± 0.19 |
| STABLE | 51.22 ± 2.12 | 57.79 ± 2.54 | 74.93 ± 0.93 | 84.08 ± 0.26 | − | − |
| SEGSL | 58.82 ± 0.76 | 61.93 ± 0.92 | 86.07 ± 0.45 | 80.12 ± 0.16 | − | − |

# E   Reproducibility

All experimental results in OpenGSL are easily reproducible. We provide more detailed information on reproducibility in the following aspects.

**Accessibility.** All the datasets, algorithm implementations, and experimental configurations can be found in our open-sourced project at `https://github.com/OpenGSL/OpenGSL` without personal request.

**Datasets.** All the datasets are publicly available. Cora, Citeseer and Pubmed are publicly available online[7], licensed under Creative Commons 4.0. BlogCatalog and Flickr were first released in [40] and further processed in [15], both to the best of our knowledge without a license. Five datasets from [34] are available in their repository[8] with the MIT license. All of the above datasets are consented by the authors for academic usage. None of these datasets contains personally identifiable information or offensive content.

**Documentation and uses.** We have made a concerted effort to offer users a documentation[9], ensuring their seamless use of our library. We have also added necessary comments to ensure code readability. In addition, we provide the necessary files to reproduce the experimental results, while also serving as examples on how to use the library[10]. To run the code, the users simply need to run `.py` files with specifies arguments such as `data, method` and `gpu`.

**License** We use an MIT license for our open-sourced project.

**Code maintenance.** We are committed to continuously updating our code while proactively addressing user issues and feedback on a regular basis. Furthermore, we enthusiastically welcome contributions from the community to enhance our library and benchmark algorithms. Nonetheless, we will implement strict version control measures to ensure reproducibility during the maintenance.

---

[7]https://linqs.org/datasets/

[8]https://github.com/yandex-research/heterophilous-graphs

[9]https://opengsl.readthedocs.io/en/latest/index.html

[10]https://github.com/OpenGSL/OpenGSL/tree/main/paper

