# OpenReview forum: "OpenGSL: A Comprehensive Benchmark for Graph Structure Learning"
_NeurIPS.cc/2023/Track/Datasets_and_Benchmarks — NeurIPS 2023 Datasets and Benchmarks Poster_

### Official Review · Reviewer_idmy · 2023-07-03

**Rating:** 6
**Confidence:** 3
**Clarity:** This paper is in general clearly writ…

**Strengths:**

1. This paper is in general clearly written and easy to follow.
2. The benchmark is performed under a controlled and fair setting to compare the performances of these baselines.
3. The observations made by this paper are interesting and may lead to future research problems. I specifically like the observations and experiments on homophily, which challenges a widely accepted assumption and sheds new lights on GSL.


**Additional Feedback:**

NA. Please see all my feedbacks above.

**Correctness:**

I would say that the experiments are done in an appropriate, controlled and fair way.

**Documentation:**

From the github link provided by the authors, I think that the documentation is OK.

**Ethics:**

No.

**Opportunities For Improvement:**

1. In Section 4.1 Performance Comparison, the authors state that "most methods cannot handle imbalanced situations". However, as an important concept, what the 'imbalance' means is not clearly stated and compared for the datasets in Table 1. Does it mean label imbalance? Anyway, it is always better to clarify and quantify what 'imbalance' means, and draw a relationship between imbalance and performance. In this way, Observation 1 can be made much stronger.

2. Some results do not seem to make sense. For example, in Table 2, GEN and ProGNN are not as competitive as naive GCN. However, in Table 4, the structure of ProGNN leads to improvements on 6/7 backbones, and GEN on 4/7. Moreover, in Table 4, column GCN, ProGNN is better than A. I feel slightly confused by the conflicting results. Please justify that.

3. It may be out of the scope of this paper, but I really would like to see several more results. The first is the consistency of the learned graph structures across multiple random seeds. The second is how similar is the learned graph structure compared to the ground truth. Finally, as some methods (e.g. ProGNN) are designed for improving robustness against structural perturbations, I would like to see results on it as well.

4. One baseline is missing, LDS (Franceschi et al. 2019). This is among the first works of graph structure learning and is cited, so please include it in the comparison.

5. Minor points.
- In Section 2, $\mathbb{R}$ should be used to represent real numbers, instead of $R$ or $\mathcal{R}$.
- In Section 7, Line 273, the authors mention Observation 7. However, there does not seem to be an observation 7 in the paper.

(Franceschi et al. 2019) Learning discrete structures for graph neural networks. ICML 2019.

**Relation To Prior Work:**

As far as I know, this is among the first benchmarks on graph structure learning.

**Summary And Contributions:**

This paper presents a benchmark for graph structure learning (GSL) methods, whose goal is to learn the graph structure along with the GNN modules. The authors compare 12 state-of-the-art GSL methods under a fair setting (i.e. datasets, data partition, backbone). The authors made several interesting findings, including that GSL does not always outperform vanilla GCN, homophily may not be a good objective, the structures learned via GSL generalize to other backbones, etc.

---

> ### Author Response · Authors · 2023-08-19
> **Rebuttal by Authors**
>
> Thanks for your valuable feedbacks. Below are our replies.
>
> **Q: Issues on imbalance**
>
> We are sorry for the confusion. "Imbalance" refers to the **imbalance in class labels**. Appx. A.1 provides a detailed description of these datasets. In these datasets, "questions" and "minesweeper" exhibit extreme class imbalance. As can be observed from Table 2, GSL is unable to handle such imbalanced datasets in homophilious graphs.
>
> **Q: Issues on seemingly contradictory results**
>
> That's a good question. One point that needs to be emphasized is that the results in Tables 2-3 are obtained from the original GSL method, where **GSL and GNN are jointly optimized**. In Tables 4-5, we treat each GSL method as a **pre-processing step** and **train a GNN model from scratch using the learned structures**. We have added an explanation about this difference in RQ3 in the revised manuscript.
>
> This observation is interesting. ProGNN does not bring improvement when optimizing the graph structure and GNN jointly, but the learned graph structure is actually a better input compared to the original structure. This observation suggests that the training procedure has a visible impact on the results. We have not thoroughly investigated into this, so it was not mentioned in the paper.
>
> **Q: It may be out of the scope of this paper, but I really would like to see several more results. The first is the consistency of the learned graph structures across multiple random seeds. The second is how similar is the learned graph structure compared to the ground truth. Finally, as some methods (e.g. ProGNN) are designed for improving robustness against structural perturbations, I would like to see results on it as well.**
>
> Regarding the first aspect, we are unsure if "multiple random seeds" refers to repeating experiments on multiple random seeds or splitting data . For the former, the first three experiments in Sec. 4 are results from 10 repeated experiments. For the latter, we used fixed splits for cora, citeseer, pubmed, blogcatalog, and flickr to align with most existing work. For the remaining five datasets we use ten different splits as in their paper.
>
> Regarding the second aspect, it's important to note that **there is no ground truth for GSL**. If "ground truth" refers to a perfect graph structure where only nodes in the same class are connected, then homophily can serve as a measure of similarity to such ground truth. We have already provided results on homophily and other measures of structure properties of the learned structure in **Sec. 4.2** and **Appx. C.1**, respectively.
>
> Regarding the third aspect, some results on robustness are already provided in **Appx. C.4**. We have added experiments with another perturbation method Metattack in the revised manuscript, and the results can also be found in Appx. C.4. From these results, it can be observed that some GSL methods exhibit better robustness compared to GNNs.
>
> **Q: One baseline is missing, LDS (Franceschi et al. 2019).**
>
> We are sorry for this omission. We have added this method in the revised manuscript and updated Tables 2-3 accordingly.
>
> **Q: Minor points.**
>
> We have carefully reviewed the paper again and corrected the errors, including the ones mentioned.

---

### Official Review · Reviewer_VFMG · 2023-07-19
**Review for OpenGSL**

**Rating:** 7
**Confidence:** 4

**Strengths:**

Besides the contributions mentioned. The paper is:
1. Well written and well-structured.
2. The implementation details are helpful to reproduce the results.
3. The code in the GitHub Repo is easy to use.
4. Documentation is well-written.
5. The insights on the homophily and the generalization are helpful to the GSL research community.

**Additional Feedback:**

N/A.

**Clarity:**

The paper is well-written and the observations are clearly stated and supported by the experiments.

**Correctness:**

Yes, all the claims sound correct to me. The evaluation methods and experiment design are appropriate and performed correctly.

**Documentation:**

Yes, the benchmark is well documented and supports reproducibility.

**Ethics:**

No, there are no ethical concerns.

**Limitations:**

I have no comments on this.

**Opportunities For Improvement:**

1. To support the observations in section 4.2, a heat map would be better to show the correlations between homophily and structure
learning.
2. The results in Tables 4 and 5 could be replaced by the decrease/increase values to show generatlizability. BTW. what is "A" in these two tables?
3. It would be better to show the performance of the evaluated methods on graphs with zero knowledge on the structure.

**Relation To Prior Work:**

Well fit and a good summarize of the prior work.

**Summary And Contributions:**

This paper presents OpenGSL, a comprehensive benchmark for Graph Structure Learning (GSL) that establishes standardized experimental
protocols in the field. A detailed overview of GSL methods and their diverse applications is provided, alongside a comprehensive description of the OpenGSL benchmark and its evaluation metrics. The paper concludes with a thought-provoking discussion on the key
findings and valuable insights derived from the application of OpenGSL to evaluate various GSL methods.

The contributions in this submission can be summarized as follows:
1. Introducing OpenGSL, the first comprehensive benchmark for Graph Structure Learning (GSL) that aims to standardize experimental protocols in the research field.
2. Implementing a wide range of GSL algorithms through unified APIs, while also adopting consistent data processing and data splitting
approaches for the purpose fair comparisons.
3. Enabling a fair comparison among twelve state-of-the-art GSL methods by unifying the experimental settings across ten popular
datasets of diverse types and scales.
4. Revealing an important observation that GSL methods do not consistently outperform the vanilla Graph Neural Networks (GNNs) on various datasets.
5. Providing multi-dimensional analysis of the GSL methods, including accuracy, computation, and memory cost.
6. Offering insights into the learned structure of the GSL methods, particularly regarding their homophily and generalizability to other GNN backbones.

---

> ### Author Response · Authors · 2023-08-19
> **Rebuttal by Authors**
>
> Thanks for your valuable feedbacks. Below are our replies.
>
> **Q: To support the observations in Sec. 4.2, a heat map would be better to show the correlations between homophily and structure learning.**
>
> In this experiment we have two analytical angles. First, we aim to study **the changes on homophily** of learned structures compared with the original structure. Second, we aim to observe **the relationship between the classification performance and the homophily**. To achieve this, we use bar charts to represent homophily (arranged from low to high, with the original structure highlighted in yellow) and line graphs to represent the classification performance. We think this way of representation can clearly reflect the patterns. We find it hard to achieve the same effect using a heatmap.
>
> **Q: The results in Tables 4 and 5 could be replaced by the decrease/increase values to show generalizability. BTW. what is "A" in these two tables?**
>
> We have made modifications to make tables in Sec. 4.3 and Appx. C.2 clearer. "A" in these two tables represents training on the original structure. We have replaced it with "Original Structure" for clarity.
>
> **Q: It would be better to show the performance of the evaluated methods on graphs with zero knowledge on the structure.**
>
> That's a great suggestion! We have added some results in the revised manuscript (See **Appx. C.7**). We replace the original structure with either an identity matrix or a KNN matrix based on the original features. The results demonstrate that GSL can still be effective on some datasets.

---

> > ### Comment · Reviewer_VFMG · 2023-08-22
> >
> > Thank you for the response.

---

### Official Review · Reviewer_cdzV · 2023-07-22
**Straightforward approach to benchmark graph structure learning methods**

**Rating:** 6
**Confidence:** 4
**Clarity:** The paper is well written and contrib…

**Strengths:**

- The proposed criteria for benchmarking the structure learning aspect are an important step towards better comparing different approaches
- GSL methodologies vary significantly, which requires elaborate criteria that capture the quality of the learned graph structure, memory requirement and time consumption. This is addressed well by using several metrics that consider the relationship between accuracy and homophily, memory and time consumption.


**Additional Feedback:**

- Maybe the structure learning aspect could be better captured via a structural perturbation approach (e.g., similar to [3] mentioned above), observing if models recover the graph structure well.
- It would be interesting to analyze more which GSL approach works best on which data, e.g., does co-training, iter-training and pre-training work better.

**Correctness:**

- The proposed way to benchmark GSL methods makes sense.
- A good aspect to study would be the capacity to scale to larger graphs.

**Documentation:**

Experimental setup and code are provided and documented well.

**Limitations:**

The considered graphs are fairly small by today's standards. The computational cost of GSL methods often prohibits the application to large graphs, which seems to be a major drawback compared to traditional GNNs. It would be a good idea to capture (and in turn encourage) scalability in a benchmark study to guide future work toward improving this aspect.

**Opportunities For Improvement:**

- Benchmarking the performance of GSL methods on the primary task/dataset seems to be closely related to benchmarking GNNs. The authors could make it more clear why it is necessary to propose separate benchmarking conventions with representative benchmarks in place [1, 2].
- Datasets like Cora, CiteSeer and PubMed are arguably suboptimal to benchmark the capacity of models to learn informative graph structure. For example, empirical analysis in [3] (Figure 6) suggests that graph structure has a limited impact on the classification performance.

[1] Dwivedi, Vijay Prakash, Chaitanya K. Joshi, Anh Tuan Luu, Thomas Laurent, Yoshua Bengio, and Xavier Bresson. "Benchmarking Graph Neural Networks." JMLR (2023).

[2] Hu, Weihua, Matthias Fey, Marinka Zitnik, Yuxiao Dong, Hongyu Ren, Bowen Liu, Michele Catasta, and Jure Leskovec. "Open graph benchmark: Datasets for machine learning on graphs." NeurIPS (2020).

[3] Liu, Renming, Semih Cantürk, Frederik Wenkel, Sarah McGuire, Xinyi Wang, Anna Little, Leslie O’Bray et al. "Taxonomy of benchmarks in graph representation learning." LoG (2022).

**Relation To Prior Work:**

Related work could be discussed more thoroughly, for example, covering other GNN benchmarking frameworks like [1].

**Summary And Contributions:**

The paper proposes a benchmark for graph structure learning (GSL) and benchmarks twelve state-of-the-art GSL methods. Their analysis considers homophily of the learned structure, the generalizability of the learned structure across GNN backbones, and the time and memory efficiency of the existing methods. They find that GSL methods do not necessarily outperform graph neural networks (GNNs) and that learning more homophile graphs does not always help model performance.

---

> ### Author Response · Authors · 2023-08-19
> **Rebuttal by Authors**
>
> Thanks for your valuable feedbacks. Below are our replies.
>
> **Q: Benchmarking the performance of GSL methods on the primary task/dataset seems to be closely related to benchmarking GNNs.**
>
> We clarify some key distinctions between OpenGSL and existing GNN benchmarks.
>
> Firstly, **we focus specifically on the GSL domain**. We make significant efforts to carefully reimplement a wide range of sota GSL algorithms. OpenGSL provides a fair evaluation of these GSL algorithms, and a convenient platform for conducting experiments. **This cannot be easily achieved using existing benchmarks**.
>
> Secondly, **we consider both homophilious and heterophilious datasets** when evaluating GSL. This is a dimension that has been largely overlooked in existing benchmarks for GNNs.
>
> Lastly, besides node classification(Sec. 4.1), **we include some experimental settings that are specific to GSL**, such as experiments on homophily/heterophily(Sec. 4.2), generalizability(Sec. 4.3), and results without original structures(Appx. C.7). These settings allow us to delve deeper into GSL, based on which we provide valuable insights and promising research directions.
>
> **Q: Datasets like Cora, CiteSeer and PubMed are arguably suboptimal to benchmark the capacity of models to learn informative graph structure.**
>
> Despite the claim in [3], we need to clarify that this only indicates that feature information **dominates**, rather than being **sufficient**, which implies that there is still room for improvement with improved structure.
>
> Moreover, [3] only indicates that the **original structure** is not informative enough. A basic motivation for GSL is that the original structure is not informative and needs refinement. Therefore **an original structure that is not informative is actually suitable for evaluating GSL**. Some works(LDS, SLAPS) have attempted GSL in scenarios where the original structure is completely missing (meaning the feature information is completely dominant). We have also added this experiment in the revised manuscript (See **Appx. C.7**).
>
> The OpenGSL is continuously being updated. We have also added results on synthetic datasets based on CSBM in the revised manuscript (See **Appx. C.5**), where we can control the information contained in structure and features. We plan to incorporate a wider range of datasets in the future to achieve more a reliable evaluation.
>
> **Q: The considered graphs are fairly small by today's standards..**
>
> We completely agree. We have already recorded the time and space consumption in Sec. 4.4 and Appx. C.3. In Sec. 5 we elaborate on this further and encourage future work towards scalable GSL.
>
> We have obtained results on the ogbn-arxiv dataset (See our Github repo), where only 4/12 GSL methods are able to run successfully. In the future, we will investigate ways to improve the efficiency of GSL methods.
>
> **Q: Related work could be discussed more thoroughly.**
>
> We have added the discussion on key differences with existing GNN benchmarks in the revised manuscript. (See **Appx. A.3**)
>
> **Q: Maybe the structure learning aspect could be better captured via a structural perturbation approach, observing if models recover the graph structure well.**
>
> We have already provided some results on robustness in **Appx. C.4**. Recently, we have added additional experiments under Metattack in the revised manuscript, and the results can also be found in Appx C.4. From these results, it can be observed that some GSL methods exhibit better robustness compared to GNNs. We have not yet explored whether the models recover the graph structure well, and we leave this interesting direction for future work.
>
> **Q: It would be interesting to analyze more which GSL approach works best on which data.**
>
> Great suggestion. Currently due to the differences in various design dimensions such as structure modeling approaches, sparsification strategies, and so on, **GSL methods with the same training procedure can have significant differences**. Therefore, we find it hard to observe the correlation between performance and this taxonomy. In the future, we plan to conduct a more fine-grained analysis from a design-choice angle.

---

### Official Review · Reviewer_Vfn3 · 2023-07-23

**Rating:** 4
**Confidence:** 5

**Strengths:**

+ Clear writing and the paper is easy to understand
+ Wide coverage of GSL works

**Additional Feedback:**

Please see my comments in previous sections.

**Clarity:**

Overall this paper is clearly written, except for the last conclusion section. I highly doubt that the conclusion is written by ChatGPT as it deviates from the original paper by large. For example, what does it mean by "renowned datasets" and "unbiased comparison"? and where does the paper "recommending innovative learning objectives"? how is "the robustness and broad applications of the acquired structure" established?

**Correctness:**

Not every experiments are grounded by concrete evidence. Some claims are too strong and even wrong.

**Documentation:**

Yes.

**Ethics:**

No dataset license is mentioned.

**Limitations:**

See my previous section for details.

**Opportunities For Improvement:**

1. My biggest concern is that the objective of graph structure learning is not clear. What the "underlying graph structure" (Line 28) means throughout the paper lacks a uniform definition. I feel like the closest definition of GSL is the optimizing the computation graph that controls the message passing in GNN models. Following this way, a clear separation between the structure learner and a static message passing model is needed. Otherwise, it remains very unclear what the learned structures could be for a certain GNN models in the experiments. Take a simple example that combines any existing GSL model with a graph attention network as the backbone, the GAT could also act as a structure learner since the attention weights control the message passing even if there exists a preceding GSL module.
2. In my opinion, this paper is far away from being a "comprehensive" benchmark as the scope of this work is very limited. Overall, only node classification is investigated and graph homophily is almost the only objective of the experiments. Note that many GSL models also target at adversarial attack/defense of the structures (e.g., Pro-GNN). Also, many sparisification techniques of the learned structures are not discussed.
3. The separation of co-training and iterative models is very confusing to me. It seems like an end-to-end GSL model could be co-training and iterative in the same time, since it is always true that the information representing graph structures in a neural network is flowed from one layer to the next.
4. The execution of experiments is flawed.
  - SLAPS is proposed for inferring a graph structure through self-supervision, and thus it is best suitable when there is no initial graph structure given. It is thus not fair to compare it against the other models where graph structures are provided a priori.
  - Not all GSL models use GCN as the backbone, e.g., Nodeformer [32] is an exception. In this case, it is also not fair to compare its performance with GCN as the performance increment will mislead readers.
  - Without a rigorous definition of GSL, it is not possible to know which learned graph structure mean in each model involved in RQ2.
5. Many claims are even wrong.
  - The first observation (Line 177) is self-contradictory, albeit no definition on the imbalanced dataset. It appears that some homophilous graphs are imbalanced, but then how could many of them works on homophilous graphs but in the same time cannot handle imbalanced situations? Or does it mean that imbalanced situations mean imbalanced training/test splits?
  - The fourth observation looks strange to me. In my opinion, homophily itself can serve as a good guidance for structure learning. Imagine an extreme case that the homophily is close to 1, then no matter how message is propagated, every node can only receive information from the neighborhood of the same class. I wonder how the homophily score is measured in Figures 2 and 3?
  - The fifth observation lacks supportive argument since only experiments on Cora and BlogCatalog are provided. If we assume Figures 2 and 3 are correct, then since BlogCatalog and Cora are the only two datasets that exhibit consistency with increased homophily and improved performance, it is yet to know whether the other datasets can still lead to strong generalizability.
6. Some missing details:
  - Tables 2 and 3: what is the time and memory limit in all experiments?
  - No definition about "imbalanced situations" is provided in RQ1. In what aspect that the dataset is imbalanced?
  - What does "On the other hand, they are already satisfied on homophilous datasets ..." mean in Line 211?

**Relation To Prior Work:**

Yes.

**Summary And Contributions:**

This paper presents a benchmark for graph structure learning. A total of three training strategies and 12 GSL models have been studied on 10 node classification datasets. A series of experiments around graph homophily have been conducted.

---

> ### Author Response · Authors · 2023-08-19
> **Rebuttal by Authors (1/3)**
>
> Thanks for your valuable feedbacks. Below are our replies.
>
> **Q1: My biggest concern is that the objective of graph structure learning is not clear. What the "underlying graph structure" means throughout the paper lacks a uniform definition.**
>
> The **objective of GSL** is to **tackle the suboptimality of the original graph structure**. The term **"underlying graph structure"** refers to **the original graph structure provided in the datasets**. The original structure may have suboptimal properties, and traditional GNNs such as GCN and SAGE heavily rely on this "underlying graph structure" for message propagation. In order to overcome this limitation, GSL is proposed to jointly optimize the graph structure and the GNN. **A key distinction of GSL** from static MPNNs is that **it learns a better graph structure** (**referred to as "learned structure" in our paper**) **for message passing**.
>
> Regarding **GAT** and similar methods that reweight the edges, **they can be considered as a special group of GSL** [1]. These methods only perform reweighting on existing edges, but are **unable to add new edges** that are not present in the original structure. As a result, their expressive power is relatively limited.
>
> Finally, while it is possible to combine certain existing GSL methods with GAT, this aspect has been rarely considered in previous works. Therefore, we do not include this in our work. Also, some other methods such as CoGSL and Nodeformer generate multiple structures. For the sake of simplicity, we exclude these methods from our experiments in Sec. 4.2 and 4.3. We have added a footnote in the revised manuscript (line 149) to clarify this. **The concept "learned structure" is clear in this setting.**
>
> **Q2: In my opinion, this paper is far away from being a "comprehensive" benchmark as the scope of this work is very limited.**
>
> We respectfully disagree with this criticism and believe that our work is comprehensive.
>
> * **Only node classification is investigated.**
>
>   We would like to emphasize two points: 1. **Most existing GSL works only use node classification for evaluation**. As the first benchmark for GSL, the goal of OpenGSL is to **achieve fair comparisons rather than improving existing works**. We believe that a detailed and fair comparison based on solely node classification is still valuable for the current GSL field. 2. **Some important works target specifically at node classification and contain designs not applicable to other tasks**, such as using labels or parameterizing the entire adjacency matrix. Extending these methods to other tasks would require significant efforts, which are beyond the scope of this paper. We agree that the GSL research community should pay more attention to tasks beyond node classification. We leave this direction for future work.
>
> * **Graph homophily is almost the only objective of the experiments. Note that many GSL models also target at adversarial attack/defense of the structures.**
>
>   The statement is not true. **In addition to** performance comparison (Sec. 4.1) and exploring graph homophily (Sec. 4.2), we also investigate the generalizability of the learned structures (Sec. 4.3) and the efficiency of GSL methods (Sec 4.4). Regarding robustness, we have **already** provided some results on robustness. We have also added more results under Metattack in the revised manuscript. Both are presented in **Appx. C.4**.
>
> * **Many sparsification techniques of the learned structures are not discussed.**
>
>   The goal of OpenGSL is to conduct fair evaluation and multi-dimensional analysis for existing GSL methods. It is beyond the scope of our work to cover design details such as sparisification techniques. Evaluating from a more fine-grained design-choice level is an interesting direction. We leave it for future work.
>
> Starting from several meaningful research questions, we conduct experiments from multiple angles and obtain valuable insights. While more work needs to be done on other tasks and design choices, we believe our work is sufficiently comprehensive as the first benchmark for GSL.
>
> **Q3: The separation of co-training and iterative models.**
>
> There seems to be a misunderstanding here. **"Iterative" does not refer to iterations in the forward process of a neural network, but the training procedure**. Co-training means the GSL module and the GNN module are **simultaneously updated under the same objective**, while the structure learning module may interleave between GNN layers in the forward process, as in the case of Nodeformer. On the other hand, for iter-training, the GSL module and the GNN module are **iteratively updated under different objectives** (e.g., GEN and SEGSL use non-gradient descent methods to learn new structures from predicted labels of a well-optimized GNN).
>
> **References:**
>
> [1] Zhu, Yanqiao, et al. "Deep graph structure learning for robust representations: A survey." *arXiv preprint arXiv:2103.03036* 14 (2021).

---

> > ### Author Response · Authors · 2023-08-19
> > **Rebuttal by Authors (2/3)**
> >
> > **Q4: The execution of experiments is flawed.**
> >
> > * **Issue on SLAPS**
> >
> >   We have added footnotes (line 117) to clarify this point. Note that the key observations we obtained in the paper would not be affected if we removed SLAPS. Also, we have added results when the graph structure is missing in the revised manuscript, presented in **Appx. C.7**.
> >
> > * **It is not fair to compare Nodeformer with GCN.**
> >
> >   In fact, **almost all methods (except Nodeformer) use GCN as the backbone**, which is why we chose GCN for fair comparisons. For Nodeformer, in their paper, **the authors compare it with GCN**, JKnet, IDGL, LDS, and other GCN-based methods, we argue it is reasonable to include Nodeformer in the comparisons.
> >
> >   Also, we have added the performance of various GSL methods using other backbones such as APPNP and GIN in the revised manuscript (see **Appx. C.5**).
> >
> > * **Without a rigorous definition of GSL, it is not possible to know which learned graph structure mean in each model involved in RQ2.**
> >
> >   Please refer to our reply in Q1.
> >
> > **Q5: Many claims are even wrong.**
> >
> > * **The first observation (Line 177) is self-contradictory.**
> >
> >   The term "imbalance" here refers to class imbalance. See **Appx. C.1** for details on the datasets. The first three datasets in Table 1 are relatively balanced, while questions and minesweeper are **highly imbalanced**. Here we intended to mean that GSL methods can be effective on **homophilous balanced** datasets but fail on **homophilous, but highly imbalanced datasets**. We have made revisions to avoid misunderstanding.
> >
> > * **The fourth observation looks strange to me. In my opinion, homophily itself can serve as a good guidance for structure learning. Imagine an extreme case that the homophily is close to 1, then no matter how message is propagated, every node can only receive information from the neighborhood of the same class. I wonder how the homophily score is measured in Figures 2 and 3?**
> >
> >   It is true that a completely homophilious graph can be effective for node classification. However, to obtain such a completely homophilious graph, one needs to know all the labels, which is impossible in reality. In fact, **the edges that can be correctly added/removed are constrained by limited supervision**. Guiding by homophily thus may not lead to a **sufficiently homophilious** structure but **confusing structural patterns**.
> >
> >   Consider a dating network where people of one gender tend to connect with people of a different gender (statistically). Guiding by homophily would add edges in the same gender and remove edges in the different. This violates the original structural pattern while fails to produce a sufficiently homophilious structure due to limited label information. This can lead to a decrease in the performance.
> >
> >   The experimental results can validate the aforementioned intuition. Please refer to Figure 2-3 (here **"homophily" refers to edge homophily in the learned structures**). In most cases, performance is not positively correlated with homophily of the learned structures, and even negatively correlated in some cases.
> >
> >   We suggest that **the objective of structure learning should consider the specific characteristics of the datasets** rather than targeting at homophily in all cases, even if it contradicts the informative original structural patterns. For the sake of rigor, we have added the word "**always**" in the revised manuscript).
> >
> > * **The fifth observation lacks supportive argument since only experiments on Cora and BlogCatalog are provided. If we assume Figures 2 and 3 are correct, then since BlogCatalog and Cora are the only two datasets that exhibit consistency with increased homophily and improved performance, it is yet to know whether the other datasets can still lead to strong generalizability.**
> >
> >   We respectfully disagree with this criticism.
> >
> >   First, in **Appx. C.2**, there are **already** experimental results about structure generalizability on other datasets. The experimental results remain **consistent** across these five datasets.
> >
> >   Second, the statement "BlogCatalog and Cora are the only two datasets that exhibit consistency with increased homophily and improved performance" is incorrect. It should be "BlogCatalog and Flickr" according to Figures 2-3. We also argue that the fifth observation is not related to this property, as similar results are observed across five datasets.

---

> > > ### Author Response · Authors · 2023-08-19
> > > **Rebuttal by Authors (3/3)**
> > >
> > > **Q6: Some missing details:**
> > >
> > > * The time limit is 24 hours for a run, and the space limit is 80GB. We have added words in Table 2-3 to clarify this .
> > > * The term "imbalanced" refers to class imbalance, as explained in the first point in Q5.
> > > * As explained in the second point in Q5, the edges that can be correctly added/removed are constrained by limited supervision. For homophilious graphs, the correct estimation of the graph structure may largely **overlap with the original structure**. And a small number of correct refinements may be difficult to outweigh the introduced noise. We suggest this may be the reason why GSL struggles to further improve the homophily on homophilious graphs.
> > >
> > > **Q7: Not every experiments are grounded by concrete evidence. Some claims are too strong and even wrong.**
> > >
> > > As stated above, we have made the necessary revisions to ensure that our statements are more rigorous. We believe that the observations in the revised version are grounded by concrete evidence.
> > >
> > > **Q8: Issues on the Conclusion**
> > >
> > > We apologize for the confusion here. We only used ChatGPT to polish our existing words but did not use it directly for summarization. We have modified "ten renowned datasets covering varying types and scopes" to "a diverse set of datasets". The phrase "recommending innovative learning objectives" was intended to recommend researchers to develop innovative learning objectives, rather than proposing them in this paper. We have modified "the robustness and broad applications" to "generalizability". We have also made several other modifications to better conclude the paper and avoid misunderstandings.
> > >
> > > **Q9: No dataset license is mentioned.**
> > >
> > > we have added the information on dataset license in the revised manuscript (See **Appx. D**).
> > >
> > >
> > >
> > > We sincerely appreciate the valuable feedbacks from the reviewer. Some concerns were raised regarding the rigor and clarity of our paper. We have provided explanations, necessary revisions and additional experiments to address them. Here we would like to highlight our contributions as follows:
> > >
> > > We achieve a fair comparison among SOTA GSL methods through careful reimplementations and unified experimental settings.
> > >
> > > Starting from several meaningful research questions, we conduct a multi-dimensional analysis and obtain valuable observations. These insights, among which the detailed analysis on homophily are especially liked, are identified helpful to GSL research community by other reviewers.
> > >
> > > As the first benchmark for GSL, OpenGSL offers a convenient platform for evaluating GSL algorithms and facilitating future research.

---

> ### Author Response · Authors · 2023-08-26
> **Requesting Feedback**
>
> Thanks once again for the efforts you spent on reviewing our work. We have carefully considered your comments and addressed each of them in our rebuttal. It would be very valuable for us to receive your feedback. If you have any further concerns, please feel free to let us know.

---

> ### Comment · Reviewer_Vfn3 · 2023-08-30
>
> Thank you for the detailed rebuttal. I have increased my score by 1. Unfortunately, while it clarifies several issues, my overall assessment remains negative primarily due to the limited scope of this study.
>
> Firstly, the claim that "Most existing GSL works only use node classification for evaluation" is inaccurate. Many highly-cited papers extend the evaluation of their GSL models beyond node classification. For example:
> * Heterogeneous node classification: Heterogeneous Graph Structure Learning for Graph Neural Networks (AAAI 2021, **130 citations**); Graph Transformer Networks (NeurIPS 2019, **651 citations**)
> * Graph classification: Hierarchical Graph Pooling with Structure Learning (AAAI 2020, **130 citations**); Graph Structure Learning with Variational Information Bottleneck (AAAI 2021)
> * Node clustering: Towards Unsupervised Deep Graph Structure Learning (WWW 2022)
> * Structure inference: NodeFormer (NeurIPS 2022); SLAPS: Self-Supervision Improves Structure Learning for Graph Neural Networks (NeurIPS 2019, 72 citations)
>
> I also do not agree that "Extending these methods to other tasks would require significant efforts". Given that OpenGSL should be modular, incorporating additional evaluations—like graph classification—should feasibly be a matter of changing the dataset loader. This should be a reasonable expectation within the one-month rebuttal period.
>
> Thirdly, the definition of graph structure learning remains ambiguous. The rebuttal mentions traditional GNNs like GCN and SAGE and their reliance on "underlying graph structure." However, GraphSAGE, for instance, drastically modifies this structure by sampling neighborhoods. This highlights the need to specify what 'structure' the GSL models are targeting. The current paper seems to pursue two intertwined goals: explaining the input graph (e.g., through the lens of homophily) and altering the message-passing paths for task-specific downstream tasks.
>
> Lastly, my concerns regarding the fourth observation have not been fully addressed. The rebuttal introduces a dating network as a heterophily example but fails to explain the unsatisfactory performance of GSL models on homophily datasets like Cora. Could the authors elaborate on what they mean by "confusing structural patterns"?

---

> > ### Author Response · Authors · 2023-08-30
> > **Response to the Remaining Concerns**
> >
> > Thanks for your valuable feedbacks. We would like to further elaborate on the remaining concerns.
> >
> > **Q: Issue on the task**
> >
> > While some use other tasks, they can be **respectively** considered as the **minority in different dimensions** and also **not suitable** for other methods. Specifically:
> >
> > * For heterogeneity, HGSL and GTN are important works specifically designed for heterogeneous graph. However, all other included GSL methods do not consider this setting, making it unfair to compare them against these two methods.
> > * For graph-level task, it has not been considered by most methods and requires additional modifications (explained later).
> > * For node clustering, it requires methods to be unsupervised, thus being not applicable to most methods. This has been discussed in Sec. 5.
> > * For structure inference, it is more of an experimental setup rather than a new task, i.e., SLAPS and Nodeformer perform node classification when the initial structure is not provided. We have provided preliminary results in Appx. C.7.
> >
> > As stated above, in **each** of the three tasks, only a **minority** of GSL methods are applicable. We aim to benchmark existing GSL methods, so we focus on node classification, which is **most shared** by existing works.
> >
> > Regarding the additional effort to extend these methods, it is **more than simply changing dataset loader**, especially for graph-level tasks. As many GSL methods are specifically designed for a single graph, extending them to graph-level tasks requires modifications to their **model design and training process**. For example, graph-level tasks do not provide node labels explicitly used by GEN and WSGNN. LDS, ProGNN, and SUBLIME involve parameterizing the entire adjacency matrix, which is challenging to implement in a graph-wise manner.
> >
> > More importantly, to fairly benchmark existing GSL works, we aim to **follow their original designs rather than improve them**. Even if it were possible to extend these methods to graph-level tasks, it would **deviate largely** from their original papers and thus be out of the scope.
> >
> > We **completely agree** that more work should be done on other tasks. We plan to investigate GSL on heterogeneous graphs, temporal graphs, as well as graph-level tasks and unsupervised tasks in the future. However, we believe that currently a fair comparison on node classification is helpful, along with other valuable insights.
> >
> >
> >
> > **Q:  Issue on the definition of GSL**
> >
> > We apologize for the confusion here. We would like to clarify that the GSL **learns** a new structure for message passing. While GraphSage also changes the graph, it is not performed **in a learnable way**. Thus it is not considered as structure learning but **still basic GNN relying on fixed structure** in current literature [1, 2, 3]. As a comparison, PTDNet [2] utilizes parametrized networks to apply dropout on edges and is therefore considered a GSL method [1, 4, 5].
> >
> > In this context, GSL either 1) **generates a structure and feeds it to subsequent GNNs relying on fixed structure** or 2) **generates the structure layer-wisely** (e.g., GAT, PTDNet, Nodeformer), both **in a learnable way**. In Sec. 4.2 and 4.3, we only considered the former for simplicity. Therefore, the definition of the learned structure in these experiments is clear.
> >
> > **Q: Issue on the fourth observation**
> >
> > We introduce the dating network as an extreme case to show that **homophily is not always a good guidance** and that **the objective of structure learning should consider the specific characteristics of the datasets**. "confusing structural patterns" means that adding/removing edges based on homophily can **disturb the existing informative structural patterns**, as seen in the example of the dating network.
> >
> > For citation datasets that are homophilious, the performances of most methods are actually satisfactory (Sec. 4.1). However, there is no positive correlation between homophily and performance on these datasets (Sec. 4.2). This further verifies the observation by showcasing that even on a homophilious graph, learning **higher homophily does not necessarily result in better performance** for different GSL methods.
> >
> > We think this observation is rigorous, as we found that homophily is only significantly correlated with performance on two datasets (Blogcatalog and Flickr). We also introduce heterophilious structural patterns, proposed in recent works, as a possible reason. These analyses are considered "interesting" and "valuable".
> >
> >
> >
> > Finally, we sincerely appreciate the time you have devoted to reviewing our paper. Your valuable feedbacks are crucial for us to improve our work. In the future, we will explore the application of GSL to other tasks and investigate what characteristics should be learned for diverse datasets beyond homophily. We hope that our response can address your remaining concerns.

---

> > > ### Author Response · Authors · 2023-08-30
> > > **References of the response**
> > >
> > > **References:**
> > >
> > > [1] Deep graph structure learning for robust representations: A survey (arXiv 2021).
> > >
> > > [2] Learning to Drop: Robust Graph Neural Network via Topological Denoising (WSDM 2021)
> > >
> > > [3] Towards Unsupervised Deep Graph Structure Learning (WWW 2022)
> > >
> > > [4] Reliable Representations Make A Stronger Defender: Unsupervised Structure Refinement for Robust GNN (KDD 2022)
> > >
> > > [5] Node Classification Beyond Homophily: Towards a General Solution (KDD 2023)

---

### Official Review · Reviewer_zkm3 · 2023-07-24

**Rating:** 7
**Confidence:** 3
**Correctness:** The experiments look sound. I did not…
**Clarity:** The paper is easy to read, and almost…

**Strengths:**

- The paper gives a comprehensive overview of the GSL field
- The authors made a great effort to reimplement SOTA GSL architectures carefully
- The evaluation pipeline and protocol look meaningful and well-designed, and the experiments seem well-executed. Results are carefully analyzed and visualized.
- I especially liked the detailed analysis with regard to homo- and heterophilic graphs
- The paper is easy to follow

In my opinion, this is a good paper that carefully analyzes the current progress and shortcomings of the current SOTA in graph structure learning, i.e., it is a valuable contribution to this research field.

**Additional Feedback:**

Not applicable.

**Documentation:**

Not applicable.

**Ethics:**

Not applicable.

**Limitations:**

Not applicable.

**Opportunities For Improvement:**

- There are a few spelling glitches, e.g., line 69 "adjacent matrix" instead of "adjacency matrix". Moreover, there are also several small typography mistakes. Hence, the paper would benefit from thorough and detailed proof reading.
- The authors only seem to have used GCN as a baseline. It would be good to consider other standard GNN baselines, such as GIN.
- The authors concentrate on node-level prediction tasks. It would have been interesting also to see some (limited) results for graph-level prediction tasks. That is, to see if the same conclusion holds in this setting compared to the node-level setting.
- The authors rely on existing datasets. Additionally, they authors could have created a synthetic dataset generator that allows for careful control over the homophily of the generated graph to conduct an even more fine-grained empirical study.

**Relation To Prior Work:**

Yes, the discussion of related work is adequate and prior work is cited when appropriate.

**Summary And Contributions:**

The paper benchmarks architectures/methods in the area of graph structure learning (GSL), i.e., optimizing the graph structure jointly with the downstream GNN model. The authors have reimplemented twelve SOTA GSL architectures and carefully empirically evaluated them on common heterophilic and homophilic benchmark datasets, guided by meaningful research questions.

---

> ### Author Response · Authors · 2023-08-19
> **Rebuttal by Authors**
>
> Thanks for your valuable feedbacks. Below are our replies.
>
> **Q: The authors only seem to have used GCN as a baseline. It would be good to consider other standard GNN baselines, such as GIN.**
>
> Most existing GSL methods use GCN as the backbone. Our goal is to evaluate whether they are effective, so we only compare them with vanilla GCN.
>
> We agree that considering more backbones would be beneficial. We have added two GNN backbones in the revised manuscript, namely APPNP and GIN, and the results are presented in **Appx. C.5**. From the results, we can see that the performance of GSL methods can be further improved with better backbones.
>
> **Q: The authors concentrate on node-level prediction tasks. It would have been interesting also to see some (limited) results for graph-level prediction tasks.**
>
> Good advice! Currently, we only consider node classification for two reasons.
>
> First, we establish OpenGSL to fairly compare existing GSL methods, so we focus on node classification, which is a **commonly used evaluation task** in this field.
>
> Second, some GSL methods **include designs that are specific to node-level tasks** (such as using node labels) and it would be challenging to adapt them to graph-level tasks.
>
> We completely agree that this field should pay more attention to tasks other than node classification, particularly graph-level tasks. We leave this interesting direction for future work.
>
> **Q: The authors rely on existing datasets. Additionally, they authors could have created a synthetic dataset generator that allows for careful control over the homophily of the generated graph to conduct an even more fine-grained empirical study.**
>
> Thanks for the advice! We have added some results on synthetic datasets in the revised manuscript, and the results can be found in **Appx. C.6**. We use the CSBM model to generate a series of datasets and observe how various GSL methods perform under different levels of homophily. From the results, an interesting observation is that some methods perform well under low homophily conditions, indicating that they are able to capture informative heterophilious graph structures generated from the CSBM model. Further exploration is needed to address real-world data, which is relatively more complex.
>
> **Q: There are a few spelling glitches, e.g., line 69 "adjacent matrix" instead of "adjacency matrix". Moreover, there are also several small typography mistakes.**
>
> We have conducted a thorough review of the paper and revise all identified errors.

---

### Author Response · Authors · 2023-08-19
**General Response**

We sincerely thank all the reviewers for the efforts on reviewing our paper and giving valuable feedbacks. We are delighted to see that our work has been recognized for its contribution to the graph structure learning community.

We appreciate reviewers for affirming our efforts to achieve a **fair** comparison through **careful reimplementations** and **unified experimental settings**, as mentioned by Reviewer zkm3, cdzV, VFMG and idmy.

Motivated by several research questions, we conduct a multi-dimensional analysis in a series of experiments, recognized as **well-designed** (Reviewer zkm3, cdzV) and **appropriate** (Reviewer VFMG, idmy). We are pleased to see that our insights are considered **valuable** (Reviewer zkm3, VFMG) and **interesting** (Reviewer idmy), with the analysis on homophily **especially liked** by Reviewer zkm3, VFMG and idmy.

Regarding our open-sourced library, we appreciate the positive feedbacks on its **usability** and **reproducibility** from Reviewer cdzV, VFMG and idmy.

We carefully considered the comments of the reviewers and we have made appropriate revisions and supplements (marked in blue in the revised manuscript), which are listed as follows:

* We have added the discussion on key differences with existing GNN benchmarks to address the concern from Reviewer cdzV. (Appx. A.3)

* We have added results under Metattack to address the concern from Reviewer Vfn3, idmy. (Appx. C.4)

* We have added results on more backbones to address the concern from Reviewer zkm3. (Appx. C.5)
* We have included a synthetic dataset based on the advice from Reviewer zkm3. (Appx. C.6)
* We have run experiments with zero knowledge on the structure based on the advice from Reviewer Vfn3, VFMG. (Appx. C.7)
* We have included LDS in the comparison, advised by Reviewer idmy.
* We have made other appropriate revisions to the words, figures and tables for better clarity.

We hope our responses will address the concerns raised by the reviewers. If you have any further concerns, please feel free to let us know.

---

> ### Author Response · Authors · 2023-08-29
> **Reminder before the end of discussions**
>
> Dear Reviewers,
>
> We have carefully considered your comments and made corresponding revisions and explanations. Since the author/reviewer discussions will close in a few days, we would like to send this friendly reminder. We want to know if our responses have addressed your concerns. If so, we kindly ask for your reconsideration of the scores.
>
> If you have any further questions, please feel free to let us know. We would greatly appreciate it if the remaining time allows us to address any remaining/new concerns you may have.
>
> Thank you so much for devoting time to making OpenGSL a better work.

---

### Comment · Area_Chair_iDEx · 2023-08-26
**Response to authors' rebuttal**

Dear reviewers,

Could you check the authors' rebuttal and other reviewers' comments? If you have further questions, please post your comments.

Please acknowledge if you have read the rebuttal and update your ratings if necessary.

Best,
AC

---

### Decision · Program_Chairs · 2023-09-22

**Decision:**

Accept (Poster)

**Comment:**

The paper presents a benchmark for graph structure learning for graph neural networks. A total of three training strategies and 12 GSL models have been studied on 10 node classification datasets. Most reviewers generally agree that the evaluation pipeline and protocol look meaningful and well-designed, and the experiments seem well-executed. The paper also offers insights into the relations between homophily and graph structure learning performance.

Some reviewers pointed out that the comparison is only made on the node classification task and that more tasks (e.g., graph classification) should be included. I also agree with this point. However, even without the comparison for the graph-level classification tasks, the paper seems still to have sufficient contributions to the field of GNNs by benchmarking the performance and providing sufficient insights on the node classification task.